# Renogrit attenuates Vancomycin-induced nephrotoxicity in human renal spheroids and in Sprague-Dawley rats by regulating kidney injury biomarkers and creatinine/urea clearance

**Acharya Balkrishna[1,2,3], Sonam Sharma[1], Vivek Gohel[1], Ankita Kumari[1], Malini Rawat[1], Madhulina Maity[1], Sandeep Sinha[1], Rishabh Dev[1], Anurag Varshney[1,2,4]\***

1 Drug Discovery and Development Division, Patanjali Research Foundation, Haridwar, Uttarakhand, India,
2 Department of Allied and Applied Sciences, University of Patanjali, Haridwar, Uttarakhand, India,
3 Patanjali Yog Peeth (UK) Trust, Glasgow, United Kingdom, 4 Special Centre for Systems Medicine, Jawaharlal Nehru University, New Delhi, India

\* anurag@patanjali.res.in

## Abstract

Vancomycin, is widely used against methicillin-resistant bacterial infections. However, Vancomycin accumulation causes nephrotoxicity which leads to an impairment in the filtration mechanisms of kidney. Traditional herbal medicines hold potential for treatment of drug-induced nephrotoxicity. Herein, we investigated protective properties of plant-based medicine Renogrit against Vancomycin-induced kidney injury. Phytometabolite analysis of Renogrit was performed by UHPLC. Spheroids formed from human proximal tubular cell (HK-2) were used for *in vitro* evaluation of Vancomycin-induced alterations in cell viability, P-gp functionality, NAG, KIM-1 levels, and mRNA expression of NGAL and MMP-7. The *in vivo* efficacy of Renogrit against Vancomycin-induced nephrotoxicity was further evaluated in Sprague-Dawley (SD) rats by measurement of BUN, serum creatinine, and their respective clearances. Moreover, eGFR, kidney-to-body weight ratio, GSH/GSSG ratio, KIM-1, NAG levels and mRNA expression of KIM-1 and osteopontin were also analyzed. Changes in histopathology of kidney and hematological parameters were also observed. Renogrit treatment led to an increase in cell viability, normalization of P-gp functionality, decrease in levels of NAG, KIM-1, and reduction in mRNA expression of NGAL and MMP-7. In Vancomycin-challenged SD rats, Renogrit treatment normalized altered kidney functions, histological, and hematological parameters. Our findings revealed that Renogrit holds a clinico-therapeutic potential for alleviating Vancomycin-associated nephrotoxicity.

## Introduction

Kidney damage (nephrotoxicity) is a serious side effect of many marketed drugs with about 19–25% of acute renal failures occurring in part by drug exposure [1]. Vancomycin, a

**Data Availability Statement:** All relevant data are within the manuscript and its Supporting Information files.

**Funding:** The author(s) received no specific funding for this work.

glycopeptide antibiotic is extensively used for the treatment of severe Gram-positive infections like Methicillin-resistant *Staphylococcus aureus* (MRSA) [2]. Patients infected with MRSA have generally been prescribed Vancomycin but its potential to cause nephrotoxicity greatly limits its clinical use [3]. About 5–35% of Vancomycin-treated patients suffer from nephrotoxicity which leads to an increase in the duration of hospitalization, costs, and the possibility of mortality [4,5]. The proximal tubules in the nephron are responsible for the regulation of blood homeostasis by regulating the levels of glucose, amino acids, water, electrolytes and bicarbonates [6]. Vancomycin induces damage in the proximal renal tubule epithelial cells present at the site of reabsorption thereby causing imbalance in the filtration and energy transport mechanisms. This damage to kidney cells is majorly mediated by Vancomycin-induced oxidative stress which leads to cell death and loss of functionality [5]. The current interventions to prevent Vancomycin-associated nephrotoxicity includes drug withdrawal, use of oral prednisone and frequent haemodialysis [7]. The Renal proximal tubular epithelial cells (RPTECs) are involved in majority of function of the proximal tubules, so in order to evaluate novel drug treatments for prevention and management of nephrotoxicity *in vitro* models using immortalized RPTECs like human kidney-2 (HK-2) cells can be utilized. Various studies have demonstrated that the 3D culture of cells induces cellular properties and functions which are more physiologically relevant to the cells *in vivo* [6,8,9]. Hence, a more detailed *in vitro* model of nephrotoxicity can be generated using cells grown in 3D culture as the apical basal polarization can then be attained in RPTECs [6]. The use of *in vivo* mammalian models can also be done for evaluation of treatments for nephrotoxicity as they allow capture of large number of clinically relevant disease markers that can be used as surrogates of human kidney injury. Vancomycin-induced nephrotoxicity studies have been previously conducted using rats wherein the histopathological changes in kidney and biochemical markers of kidney injury obtained from serum and urine have been utilized to screen suitable therapeutic interventions [10]. Various studies have shown that free radicals generated due to Vancomycin treatment cause alterations in the antioxidant defence mechanism leading to injury in the tubular epithelial cells [11]. Herbal extracts are known to be used for the treatment of Vancomycin associated nephrotoxicity [12]. Renogrit, an ayurvedic herbal medicine is composed from several extracts of reno-protectant herbs as mentioned in the Ayurvedic text, Bhavprakash Nighantu (Table 1). The current study evaluated the potency of Renogrit in 3D *in vitro* and *in vivo* rat model of Vancomycin associated nephrotoxicity using various markers of kidney injury like cytotoxicity, loss of transporter functionality, accumulation and clearance of waste products, oxidative stress and histopathology. Clinical biomarkers of renal tubular injury namely Kidney injury molecule-1 (KIM-1), N-acetyl-β-D-glucosidase (NAG), and Neutrophil gelatinase-associated lipocalin (NGAL) [13–15] were also evaluated.

**Table 1. Herbal composition of Renogrit.**

| Scientific name | Sanskrit names | Vernacular name | Parts used | Reference Book | Page No. |
|---|---|---|---|---|---|
| *Achyranthes aspera* L. | (अपामार्गकः खरमञ्जिकः) (**Apāmārgakaḥ kharamañjikaḥ**) | Apamarg | Root | B.P.N | 400–401 |
| *Saxifraga ligulata* Murray | (सूचीतन्तुकः तन्तुलः) (**Sūcītantukaḥ tantulaḥ**) | Pashanbhed | Root | B.P.N | 101–102 |
| *Butea frondosa* Roxb. ex Willd. | (पलाशकः रक्तपुष्पः) (**Palāśakaḥ raktapuṣpaḥ**) | Palash | Flower | B.P.N | 524–525 |
| *Crateva nurvala* Buch.-Ham. | (वरुणकः सतिपुष्पः) (**Varuṇakaḥ sitapuṣpaḥ**) | Varun | Bark | B.P.N | 531 |
| *Boerhavia diffusa* L. | (पुनर्नवकः रक्तकाण्डः) (**Punarnavakaḥ raktakāṇḍaḥ**) | Punarnavamool | Root | B.P.N | 408–409 |
| *Cichorium intybus* L. | (कासनिका ग्राम्या) (**Kāsanikā grāmyā**) | Kasni | Whole plant | B.P.N | 796 |
| *Cichorium intybus* L. | Same as above | Kasni | Seed | B.P.N | 796 |
| *Tribulus terrestris* L. | (गोक्षुरकः तुरकिण्टः) (**Gokṣurakaḥ trikaṇṭaḥ**) | Gokharu | Fruit | B.P.N | 279–281 |

B.P.N- Bhavprakash Nighantu.

## Materials and methods

### Reagents

Renogrit (Laboratory internal batch # D4/CHM/SOLE169/0622) was sourced from Divya Pharmacy, Haridwar, India. Standards for UHPLC analysis namely Bergenin, Methyl gallate were obtained from TCI chemicals, India; Gallic acid from Sigma-Aldrich, USA; Quercetin from SRL, India; and Boeravinone B from Natural remedies, India. Dulbecco's Modified Eagle Medium (DMEM), Nutrient Mixture F-12 Ham, and antibiotic-antimycotic solution were obtained from Sigma-Aldrich, USA. The Nunclon Sphera U-shaped-bottom 96-well microplate, PrestoBlue, TRIzol, Verso cDNA synthesis kit, and PowerUp SYBR Green Master Mix were procured from Thermo Fisher Scientific, USA. Heat-inactivated FBS and p-Nitrophenol were obtained from HiMedia, India. Vancomycin (Lot # VCD-002B) was bought from Elisun Biotech, India. Calcein AM was obtained from Cayman Chemical, USA. Cilastatin was acquired from TCI chemicals, India. N-acetyl-L-Cysteine (NAC) and 4-Nitrophenyl-N-acetyl-β-D-glucosaminide (NAG substrate) were obtained from SRL, India. Human KIM-1 ELISA kit was obtained from GBiosciences, USA. The rat KIM-1 ELISA kit was obtained from Cusabio, China.

### Phytochemical analysis of Renogrit

Renogrit (500 mg) powder was diluted with 10 mL solution of water: methanol (70:30) and sonicated for 30 min, centrifuged at 10,000 rpm for 5 min by Sorvall ST-8R (Thermo Fisher Scientific, USA) and filtered using 0.45 μm nylon filter. Quantitative analysis of metabolites was performed by Prominence-XR UHPLC system (Shimadzu, Japan) equipped with Quaternary pump (NexeraXR LC-20AD XR), diode array detector (DAD SPD-M20 A), Auto-sampler (Nexera XR SIL-20 AC XR), Degassing unit (DGU-20A 5R) and Column oven (CTO-10 AS VP). Separation was achieved using a Shodex C18-4E (5 μm, 4.6 × 250 mm) column subjected to binary gradient elution. The two solvents used for the analysis consisted of water containing 0.1% acetic acid (solvent A) and acetonitrile (solvent B). Gradient programming of the solvent system was initially at 0–5% B for 0–10 min, 5–10% B from 10–20 min, 10–20% B from 20–30 min, 20–30% B from 30–40 min, 30–50% B from 40–50 min, 50–70% B from 50–60 min, 70–90% B from 60–70 min, 90–0% B from 70–72 min, 0% B from 72–75 min with a flow rate of 1 mL/min. 10 μL of standard and test solution were injected, column oven temperature was maintained at 35°C and detector wavelength was set 270 nm throughout the analysis. Furthermore, the identification of other major metabolites was performed by ultra-performance liquid chromatography coupled with electrospray ionization quadrupole time of flight tandem mass spectrometry (UPLC/ MS QToF). The instrumentation setup comprised of Xevo G2 XS QToF mass spectrometer (Waters Corporations, USA) connected to the ACQUITY UPLC I Class System via electrospray ionization (ESI) interface in negative mode of ionization. The capillary voltage, cone voltage, source temperature and desolvation temperature were maintained at 2.0 kv, 40 V, 120°C and 500°C, respectively. High purity nitrogen gas was used for desolvation and cone, with gas flow rates 900 and 50 Lh-1. The low collision energy (low CE) of 6 eV and high collision energy (High CE) of 15–50 eV were applied in the collision cell. To ensure mass accuracy of the optimized MS conditions, leucine enkephalin (m/z 554.2620 in negative mode) was used as a reference (lock mass) at a concentration of 200 pg/mL and a flow rate of 10 μL/min. Chromatographic separations were achieved using an ACQUITY UPLC HSS T3 (Waters Corporation, USA) (100 × 2.1 mm, 1.8 μm) column. The column temperature was maintained at 35°C throughout the analysis, whereas samples were kept at 15°C for analysis. The elution was carried out at a flow rate of 0.3 mL/min using gradient elution of mobile phase 0.1%

formic acid in water (mobile phase A) and 0.1% formic acid in acetonitrile (mobile phase B). The elution gradient program of mobile phase B was, 2% for 0 to 5 min, 2 to 5% for 5 to 10 min, 5 to 20% for 10 to 30 min, 20 to 60% for 30 to 50 min, 60 to 80% for 50 to 60 min, 80 to 2% for 60 to 61 min, 2% for 61 to 65 min. 1 μL of the sample solution was injected for scanning and the chromatogram was recorded.

## Spheroid culture

HK-2 cells were obtained from ATCC, USA. The cells were grown with DMEM/F-12K media containing 5% FBS. Cells from passage 3–5 were used for generation of spheroids post 80% cell confluency. In order to determine the optimum number of cells for generation of robust spheroids, the cells number was analyzed from 300–20,000 cells per well suspended in 200 μL media in a Nunclon Sphera U-shaped-bottom 96-well microplate. The plates were then centrifuged at 1500 rpm for 10 min and incubated at 37˚C with 5% $CO_2$ for 16 days with media change after every 3 days. Images of spheroid development were acquired after every 4 days using a Zeiss Primovert bright-field microscope (Carl-Zeiss, Germany). Further experiments were performed using HK-2 spheroids with initial seeding density of 10,000 cells per well. The spheroids were grown for 9 days with media change every 3 days after which 100 μL of the media was removed and 2× concentration of drugs was added after 2 subsequent media changes. The plates were than incubated for 5 days after which the spheroids were analyzed for different renal tubular injury markers. Untreated control (UC) was used as the normal control and NAC (4 mM) was used as the positive control.

## Cell viability

The 9-day old HK-2 spheroids were treated with a final concentration of Vancomycin (0.5–8 mM); Renogrit (10, 30, and 100 μg/mL); or co-treated with Vancomycin (2 mM) and Renogrit (10, 30, and 100 μg/mL) or NAC (4 mM) for 5 days. After 5 days, PrestoBlue (20 μL) cell viability reagent was added to the media and the plates were read for fluorescence at Ex.560/ Em.583 nm by Envision multimode plate reader (PerkinElmer, USA). Data were presented as mean ± SEM (n = 3).

## P-glycoprotein (P-gp) function analysis

P-gp efflux analysis was done as per the protocol by Im, Dai Sig *et al* [16]. HK-2 spheroids were co-treated with Vancomycin (2 mM) and Renogrit (10, 30, and 100 μg/mL) or NAC (4 mM) or Cilastatin (200 μg/mL) and incubated for 5 days. Post incubation spheroids were washed thrice by subsequent media changes and incubated with 0.5 μM Calcein AM reagent for 30 min. The spheroids were washed with DPBS and in the final wash the spheroids were broken with rigorous pipetting. The plates were read for fluorescence at Ex.494/ Em.517 nm by Envision multimode plate reader (PerkinElmer, USA). Data were presented as mean ± SEM (n = 3).

## NAG evaluation

The HK-2 spheroids were treated with Vancomycin (2 mM) and Renogrit (10, 30, and 100 μg/ mL) or NAC (4 mM) and incubated for 5 days. Post incubation 100 μL supernatant from each treatment group was collected and centrifuged at 3000 rpm for 10 min to remove cell debris. Assay was started by taking 20 μL of samples into two separate wells as sample and sample blank in flat transparent 96-well plate along with standards of p-Nitrophenol (28 μM-750 μM). Reaction was stopped in sample blank wells by adding 100 μL of stop solution (Glycine buffer,

10.6 pH). Next, 80 μL NAG substrate solution (2.2 mM, pH 4.4) was added in standard, sample, and sample blank wells. Plates were incubated at 37˚C for 2 hr. Finally, enzymatic reaction was stopped in sample and standard wells by changing the pH with stop solution, and optical density was measured at 405 nm using Infinite 200Pro plate reader (Tecan, Switzerland). NAG activity (U/L) was evaluated by using following equation:

$$\text{NAG Activity} = \frac{\text{Sample (OD)} - \text{Sample blank (OD)}}{t*\text{slope}} * \text{D.F} \left(\frac{U}{L}\right)$$

Where, Sample (OD) and Sample blank (OD) is the absorbance for each sample and its blank respectively. Slope value was taken from the linear regression equation of p-Nitrophenol curve, t is the reaction time (120 min), and D.F is the dilution factor. Data were presented as mean ± SEM (n = 3).

## KIM-1 assessment

After 5 days' treatment as mentioned before, 6 spheroids per treatment were pooled, centrifuged at 5000 rpm for 10 min, washed and centrifuged again. 300 μL of PBS was added to the spheroid pellet and the cells were kept in liquid nitrogen and lysed by 3 consecutive freeze thaw cycles. The sample were centrifuged at 14,000 rpm for 15 min and the lysate was used for performing KIM-1 ELISA (GBiosciences, USA) as per the manufacturer's instructions. Data were presented as mean ± SEM (n = 6).

## Gene expression analysis

Total RNA was extracted from HK-2 spheroids using TRIzol method according to manufacturer's instructions. 1 μg of total RNA was used to synthesized cDNA using Verso cDNA synthesis kit. Real-Time PCR (qTOWER3G machine, Analytik-Jena, Germany) was used to detect the mRNA expression level of NGAL, MMP7, using PPIA as an endogenous control. The primer sequences for each gene are as follows: NGAL F-GAAGTGTGACTACTGGATCAGGA, R-ACCACTCGGACGAGGTAACT; MMP7 F- ATGTGGAGTGCCAGATGTTGC, R-AGCAGTT CCCCATACAACTTTC, and PPIA F-CCCACCGTGTTCTTCGACATT; R- GGACCCGTATGC TTTAGGATGA. Each qRT- PCR reaction consisted of 2 μL (10 ng) of template cDNA, 0.5 μL (200 nM) each of forward and reverse primers, 5 μL of PowerUp SYBR Green Master Mix and 2 μL of ddH$_2$O. The qRT-PCR reaction was carried out under the following conditions: 95˚C for 5 min, followed by 40 cycles of amplification, denaturation at 95˚C for 30 sec, annealing at 60˚C for 30 sec, and elongation at 72˚C for 30 sec with a melt curve of 60˚C- 95˚C. The obtained C$_t$ value was normalized with an endogenous control and quantification of the samples was calculated by the $2^{-\Delta\Delta CT}$ method. Data were presented as mean ± SEM (n = 3).

## Ethics statement for *In vivo* nephroprotective assessment

The current study is reported in accordance with ARRIVE guidelines [17]. The Institutional Animal Ethics Committee of Patanjali Research Institute reviewed the proposed animal experimental protocol and subsequently approved these experiments, vide approval number PRIAS/LAF/IAEC-132. All husbandry practices and procedures were conducted under strict conformance with the standards prescribed by Committee for Control and Supervision of Experiments on Animals (CCSEA), Department of Animal Husbandry and Dairying, Ministry of Fisheries, Animal Husbandry and Dairying, Government of India.

## Experimental animals

The study was conducted on specific-pathogen-free, male Sprague Dawley rats, purchased from Hylasco Bio-Technology Pvt. Ltd., Telangana, India, which is a Charles River Laboratories-licensed animal supplier. During the entire duration of the study, the animals received gamma irradiated, standard pelleted diet (Purina 5L79 Rodent Lab diet, USA) and reverse osmosis purified drinking water, *ad libitum*, in a registered animal house (Registration # 1964/PO/RC/S/17/CPCSEA). The temperature maintained in the animal rooms was 23 ± 2˚C; and the relative humidity ranged from 40 to 60%. The photoperiod adhered to was a 12-hour light-dark cycle.

The rat equivalent doses of Renogrit were computed based on the body surface area of the animals. The recommended human dose of the Renogrit is 2000 mg/day. Animal equivalent doses (mg/kg) were calculated by multiplying human equivalent dose (33.33 mg/kg/day) by factor of 6.2 [18]. Resultant therapeutic equivalent doses for rats were determined to be 207 mg/kg/day. Rounding off to the nearest hundred, 200 mg/kg/day or 100 mg/kg b.i.d. was considered as the rat equivalent dose, corresponding to the prescribed human dose and is designated as the therapeutic dose (TD). The remaining doses employed in the in-vivo experiments were 20 mg/kg ($1/10^{th}$ of TD), 60 mg/kg/day ($1/3^{rd}$ of TD) and 600 mg/kg/day (3 times the TD).

## Establishment of the kidney injury model and compound administration regimen

Kidney injury was induced by systemic exposure to Vancomycin [5,19,20]. Briefly, after completion of a one-week quarantine period, rats were transferred to an experimental room, randomized based on their body weights and allocated to six groups:

Group 1: Normal control group
Group 2: Disease control group
Group 3: Renogrit- 20 mg/kg (mpk)/day treated group
Group 4: Renogrit- 60 mpk/day treated group
Group 5: Renogrit- 200 mpk/day treated group
Group 6: Renogrit- 600 mpk/day treated group

Animals allocated to Groups 3–6 received Renogrit at incremental doses prophylactically by oral gavage, twice a day, 14 days prior to disease induction. On the other hand, animals designated to Groups 1 and 2 received 0.5% sodium carboxymethyl cellulose, which was employed as the vehicle for formulating Renogrit. After completion of prophylactic administration, animals allotted to Groups G2-G6 received Vancomycin injection by intraperitoneal route at the dose of 200 mpk in the morning and 150 mpk in the night, at 12 hr intervals. Vancomycin administration was continued for seven consecutive days. Animals of Group 1 were administered an equal volume of sterile water, by intraperitoneal route using an identical disease induction regimen. Vehicle/compound administration was continued throughout the disease induction procedure. On Day 8, animals were administered drinking water (2 mL/100 g) transferred to metabolic cages (Orchid Scientific, India) for the collection of urine for a period of 24 hr. Water loading was additionally repeated at 4 and 8 hr.

## Urine collection and processing for biochemical parameters

On day 9, the animals were removed from the metabolic cages and the urine volume was recorded. Subsequently the urine was subjected to clinical chemistry analysis, wherein the supernatant of urine obtained by centrifugation at 800×g for 15 min at 4˚C, was evaluated for the levels of creatinine (CREAT) and urine urea nitrogen (UUN). The NAG evaluation was

performed as discussed before. The analysis of rat KIM-1 was done by sandwich ELISA (Cusabio, China) as per the manufacturer's instructions. The values of KIM-1 were further normalized with urinary creatinine. Data were presented as mean ± SEM (n = 6).

### Hematology and serum clinical chemistry analysis

After completion of urine collection, the animals were transiently anaesthetized with isoflurane, blood was collected from the retro-orbital plexus and dispensed in two centrifugation tubes. One tube contained ethylenediaminetetraacetic acid dipotassium salt for estimation of hematological parameters, whereas the other tube devoid of any anticoagulant was used for separation of serum for the estimation of clinical chemistry parameters. For enumeration of the complete blood count, blood was aspirated in BC5000Vet, a 5-part hematology analyzer qualified for veterinary purpose (Mindray, China). Data were presented as mean ± SEM (n = 6).

Blood obtained for the estimation of clinical chemistry parameters was centrifuged at $2000 \times g$ for 15 min at 4˚C following which the serum was separated Further, the sera were processed by utilizing Erba EM-200 clinical chemistry analyser (Transasia, India) for the estimation of CREAT, blood urea nitrogen (BUN), aspartate transaminase (AST) and alanine transaminase (ALT) respectively. Additionally, the sodium ($Na^+$), potassium ($K^+$) and calcium ($Ca^{2+}$) levels were also evaluated in the serum by utilizing ST-200 Plus electrolyte analyzer (Sensacore Medical Instrumentation, India). Data were presented as mean ± SEM (n = 6). The creatinine clearance, BUN clearance and the estimated glomerular filtration rate were calculated as reported by Pestel et al., 2007 [21].

### Animal necropsy and weighing of the kidney

Animals were sacrificed by intraperitoneal injection of thiopentone sodium (150 mg/kg). Subsequently, the bilateral kidneys were excised and were weighed. The weight of the kidney was expressed as a percentage of the body weight of the animal and was calculated by using the following formula:

$$\text{Relative kidney weight (\%)} = \frac{\text{Weight of the kidney (g)}}{\text{Terminal body weight (g)}} \times 100$$

### Processing of the kidneys for oxidative stress markers and gene expression

The right kidney was sectioned into two halves of which, one portion was snap frozen in liquid nitrogen for biochemical evaluations whereas the other one was dispensed into tubes containing RNAprotect tissue reagent (Qiagen, Germany) for the ensuing gene expression analysis. The samples for biochemical and molecular evaluations were stored immediately at -80˚C.

### GSH/GSSG assay in the kidney tissue

The level of GSH and GSSG were measured as per the protocol of Hissin & Hilf [22]. Briefly, the samples were incubated with phosphate buffer at pH 8 and 12 respectively and then incubated with O-pthaldehyde (OPT) for GSH detection and with both OPT and N-ethylmaleimide (NEM) for GSSG detection. All the fluorescence measurements were recorded through Envision Microplate Reader (PerkinElmer, USA) at Ex.350/Em.420 nm. Data were presented as mean ± SEM (n = 6).

### Quantitative Real-time PCR for mRNA expression in rat kidney

The total RNA was isolated from the kidney using TRIzol reagent and RNeasy mini kit according to the manufacturer's instructions. Total RNA (1 μg) was reverse-transcribed to cDNA using the Verso cDNA synthesis kit. The prepared cDNA was stored at -20°C until further use. For qRT-PCR, 10 μL of qRT-PCR reaction mixture containing 2 μL of template cDNA (5 ng), 0.5 μL (200 nM) each of forward and reverse primers: KIM-1 F- `TGGCACTGTGACATCCT CAGA,` R-`GCAACGGACATGCCAACATA`; Osteopontin F-`CCGATGAGGCTATCAAGGTC`, R-`AC TGCTCCAGGCTGTGTGTT`, and GAPDH F- `TGTGAACGGATTTGGCCGTA`; R-`TGAACTTG CCGTGGGTAGAG`. 2 μl of RNAse-free water and 5 μL of PowerUp SYBR Green Master Mix were used. The reaction was carried out under the following conditions: 95°C for 5 min followed by 40 cycles of denaturation (95°C for 30 sec), annealing (60°C for 30 sec), and extension (72°C for 30 sec) and final extension cycle at 72°C for 5 min. The intensity of fluorescence was captured at each cycle using a qTOWER3G Real-Time System Machine (Analytik Jena, Germany). GAPDH was used as a housekeeping gene and fold changes in relative mRNA expression were assessed from threshold ($C_T$) values using the $2^{-\Delta\Delta CT}$ method. Data were presented as mean ± SEM (n = 6).

### Histopathology

The left kidney was longitudinally sectioned and was stored in 10% neutral buffered formalin. The kidney was then subjected to processing by employing standard procedures in TP 1020 tissue processor (Leica Biosystems, India). Then they were embedded in paraffin wax by utilizing Histocore Arcadia H-C embedding station (Leica Biosystems, India). From the obtained tissue blocks, sections of 3–5 μm thickness were prepared by using RM 2245 microtome (Leica Biosystems, India). The sections were transferred to clean, grease-free slides, deparaffinized, and stained with hematoxylin-eosin. The stained sections were then examined microscopically by using a AxioScope-A1 microscope (Carl Zeiss, Germany) and imaging was performed using Axiovision software Version 4.9.1 (Carl Zeiss, Germany). The pathological lesions of acute kidney injury were assessed through different parameters namely A. Tubular necrosis, B. Tubular dilatation, C. Interstitial inflammation and D. Tubular cast. The severity of these lesions was assigned as 0 = absent, 1 = minimal, 2 = mild, 3 = moderate and 4 = severe [19]. If any severity was in between two grades, then additional 0.5 score was added to the lower grade. Finally, the summation of all these four parameters was assigned as total lesion score. Data were presented as mean ± SEM (n = 6).

### Statistical analysis

Data for the investigated parameters were compiled from the study groups and were then expressed as mean ± SEM. The statistical analysis was conducted by utilizing GraphPad Prism version 7.04 software (San Diego, USA). A one-way analysis of variance (ANOVA), which was followed by Dunnett's multiple comparison post-hoc test was employed to compute the statistical differences between the mean values. A p value < 0.05 was considered to be statistically significant.

## Results

### Phytometabolite analysis of Renogrit

The qualitative and quantitative analysis of Renogrit via UHPLC and MS-QToF revealed the presence of Gallic acid (CAS # 149-91-7), Bergenin (CAS # 477-90-7), Methyl gallate (CAS # 99-24-1), Quercetin (CAS # 117-39-5), Boeravinone B (CAS # 114567-34-9), Butrin (CAS #

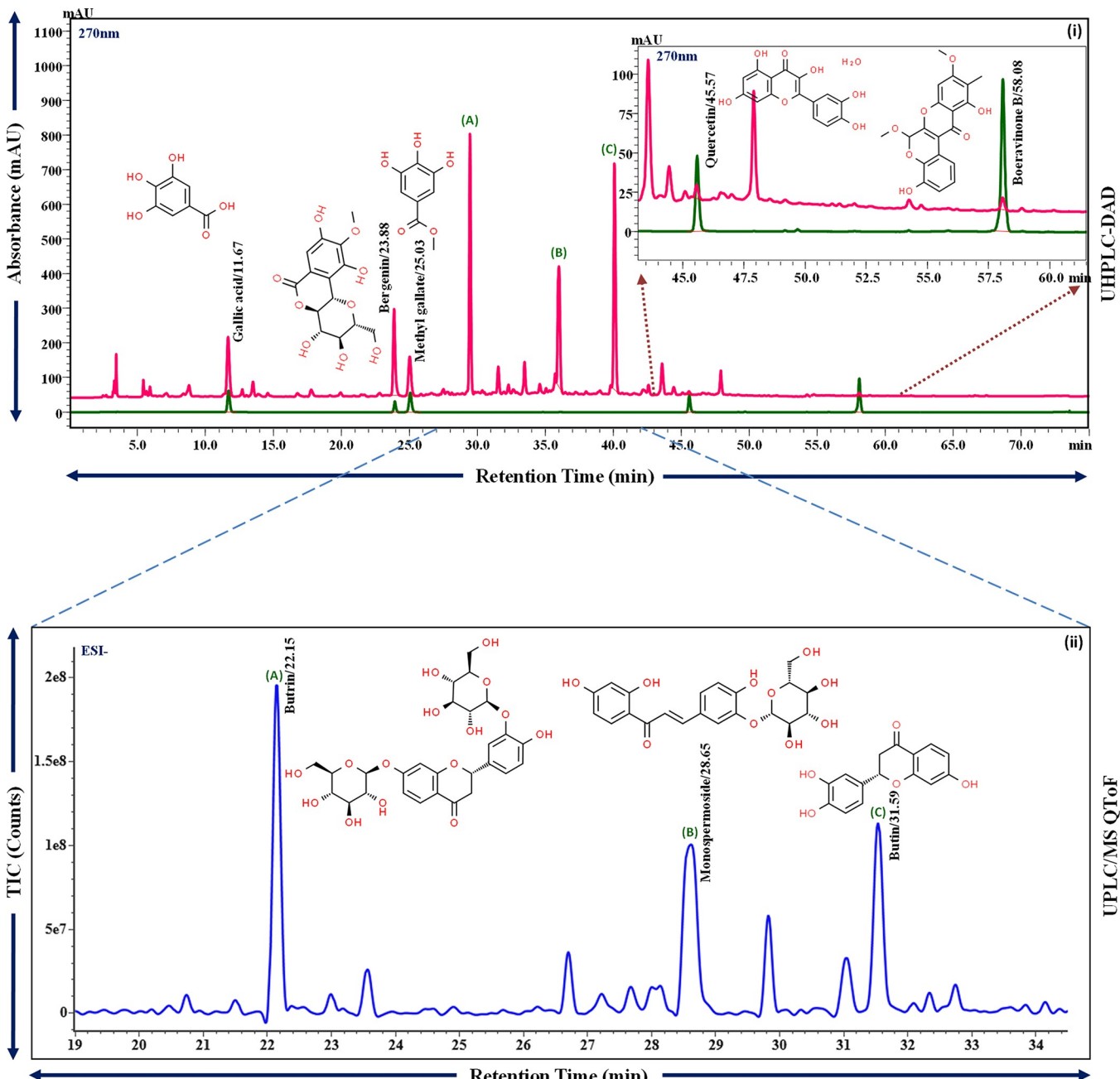

**Fig 1. Phytochemical profile of Renogrit. (i)** UHPLC-DAD chromatogram of Renogrit (Red lines) at 270 nm compared with reference standard mix (Green lines) includes, Gallic acid, Bergenin, Methyl gallate, Quercetin, Boeravinone B. **(ii)** The major peaks, found in UHPLC-DAD chromatogram, named (A), (B) and (C) were characterized as Butrin, Monospermoside and Butin respectively by UPLC/MS QToF in negative mode of ionization, Fig (ii) (Blue line).

487-52-5), Monospermoside (CAS # 30382-19-5) and Butin (CAS # 492-14-8) (Fig 1). Using reference standards, the metabolites were quantified on UHPLC as, Gallic acid (3.03 μg/mg), Bergenin (8.46 μg/mg), Methyl gallate (2.19 μg/mg), Quercetin (0.15 μg/mg), and Boeravinone B (0.06 μg/mg) (Fig 1, i).

## Development of HK2 spheroids and cell viability assessment

Cell density of 10,000 cells/well was observed to be optimum for the development of homogeneously sized spheroids which were able to remain intact after several cycles of media changes up to day 16 (Fig 2A). Vancomycin (2 mM) was selected for induction of toxicity in HK-2 spheroids (S1A Fig) as at higher concentrations satellite colonies started to form around spheroids which might lead to confounding bias in our results. Renogrit (10, 30, and 100 μg/mL) treatment did not show any toxicity in spheroids (S1B Fig). Co-treatment of Vancomycin (2 mM) with Renogrit (10–100 μg/mL) showed a significant ($p < 0.05$) increase in the % viability of HK-2 spheroids. A significant ($p < 0.01$) increase in % viability was also observed in NAC (4 mM) treated spheroids (Fig 2B). Hence, Renogrit was found to decrease the Vancomycin-induced toxicity in the 3D culture of RPTECs.

## Renogrit attenuated Vancomycin-induced suppression of P-gp function in HK-2 spheroids

A decrease in expression and function of P-gp in RPTECs is one of the major mechanisms of Vancomycin associated nephrotoxicity [16]. We investigated the effect of Renogrit (10, 30, and 100 μg/mL) on functionality of P-gp in Vancomycin (2 mM) stimulated HK-2 spheroids. It was observed that Renogrit was able to normalize the P-gp based transport function of the cell based on the fluorescence of the accumulated Calcein, a known P-gp substrate [23]. Spheroids treated with Vancomycin (2 mM) showed a significant ($p < 0.001$) increase in fluorescence which depicts that P-gp transport function has been suppressed in presence of Vancomycin. But in the Renogrit (10, 30, and 100 μg/mL) co-treated groups, the P-gp functionality was normalized as observed from the significant decrease ($p < 0.001$) in the % fluorescence (Fig 2C). Similar results were also observed in presence of the positive control drug used in this assay, Cilastatin (200 μg/mL), which is known to attenuate Vancomycin-suppressed P-gp in HK-2 cells [16].

## Renogrit reduced kidney injury markers in Vancomycin induced HK-2 spheroids

A significant ($p < 0.001$) increase in the levels of NAG (U/L) and KIM-1 (pg/mL) was observed in the Vancomycin (2 mM) treated spheroids. NAG and KIM-1 are known biomarkers of nephrotoxicity and are also clinically used for the assessment of the level of kidney injury [24]. Renogrit (10, 30, and 100 μg/mL) treatment significantly ($p < 0.001$) decreased the Vancomycin-induced injury as observed from the normalized levels of both NAG and KIM-1. Similar results were observed in NAC (4 mM) treated spheroids (Fig 2D and 2E). Hence, Renogrit might be effective in decreasing Vancomycin-induced kidney injury as evident from the improved levels of the nephrotoxicity biomarkers.

## Renogrit modulated genes expression in Vancomycin-induced HK-2 spheroids

In normal physiological conditions NGAL is expressed at very low levels in kidney but in instances of nephrotoxic injury its expression levels are substantially increased especially in the proliferating proximal tubular cells [15]. Matrix metalloproteinase-7 (MMP-7) is another major predictor of renal tubular injury [25–27]. A significant ($p < 0.001$) rise in the NGAL and MMP-7 gene expression levels was observed post Vancomycin (2 mM) induction which was found to significantly ($p < 0.05$) decreased in Renogrit (10, 30, and 100 μg/mL) or NAC (4 mM) co-treated HK-2 spheroids (Fig 2F and 2G).

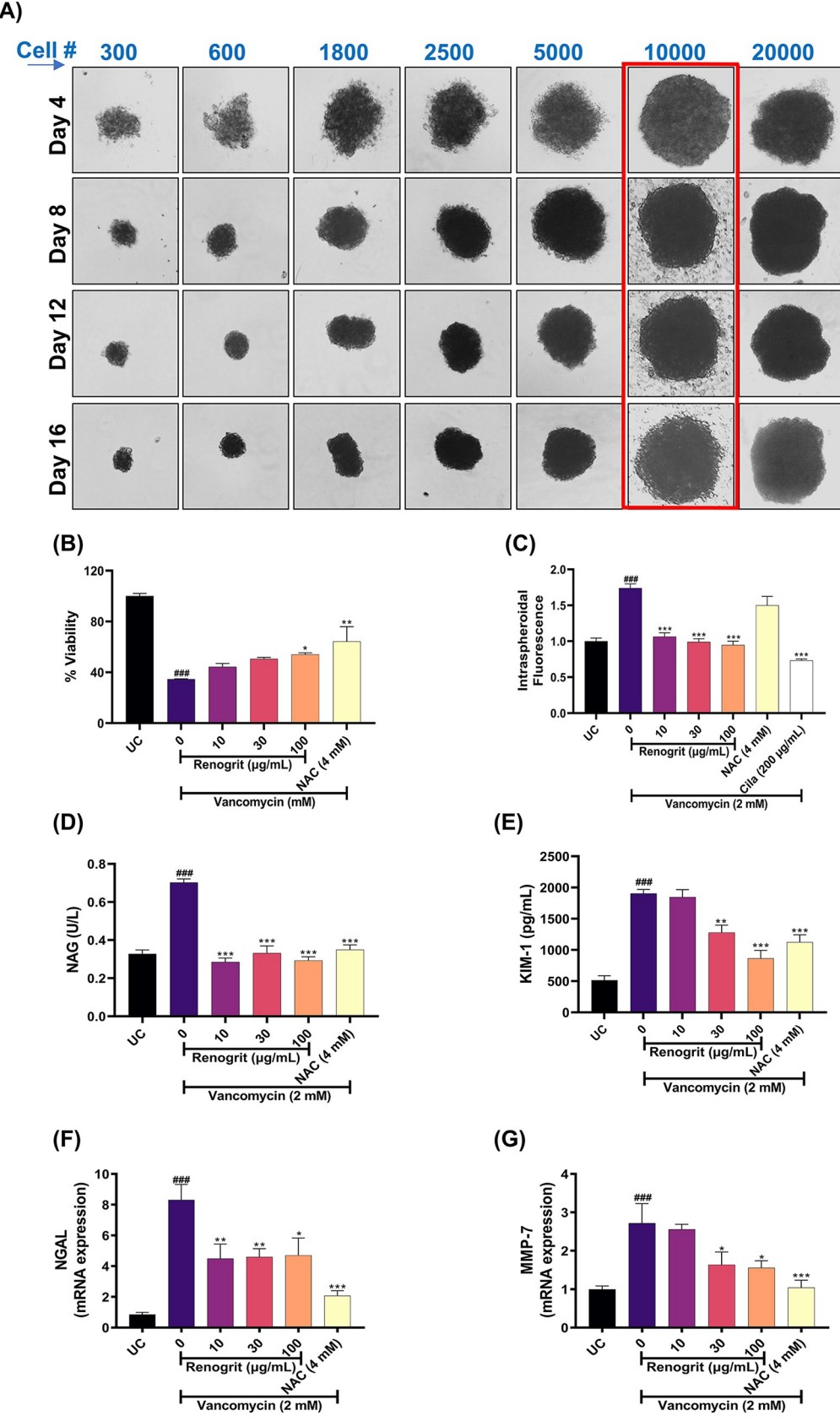

**(A)**

**(B)** % Viability — UC, Vancomycin (mM): 0, Renogrit (µg/mL): 10, 30, 100, NAC (4 mM)

**(C)** Intraspheroidal Fluorescence — UC, 0, Renogrit (µg/mL): 10, 30, 100, NAC (4 mM), Cila (200 µg/mL), Vancomycin (2 mM)

**(D)** NAG (U/L) — UC, 0, Renogrit (µg/mL): 10, 30, 100, NAC (4 mM), Vancomycin (2 mM)

**(E)** KIM-1 (pg/mL) — UC, 0, Renogrit (µg/mL): 10, 30, 100, NAC (4 mM), Vancomycin (2 mM)

**(F)** NGAL (mRNA expression) — UC, 0, Renogrit (µg/mL): 10, 30, 100, NAC (4 mM), Vancomycin (2 mM)

**(G)** MMP-7 (mRNA expression) — UC, 0, Renogrit (µg/mL): 10, 30, 100, NAC (4 mM), Vancomycin (2 mM)

**Fig 2. Development of HK2 spheroid *in vitro* culture model and evaluation of kidney injury biomarkers. (A)** Representative brightfield image of day 4 to day 16 old HK2 spheroid with different cell densities (300–20,000 cells/ well). Spheroids with cell density of 10,000 cells/well and diameter between 300–350 μm were used for all experiments. **(B)** Viability analysis of HK2 spheroids by PrestoBlue dye post co-treatment of Vancomycin (2 mM) and Renogrit (10, 30, and 100 μg/mL) or NAC (4 mM). **(C)** P-gp function analysis by Calcein-AM (0.5 μM) reagent on Vancomycin (2 mM) and Renogrit (10, 30, and 100 μg/mL) or NAC (4 mM) co-treated HK2 spheroids by evaluation of difference in fold of fluorescence intensity. Renogrit treatment decreased the Vancomycin (2 mM) stimulated release of nephrotoxicity biomarkers, **(D)** NAG (U/L) and **(E)** KIM-1 (pg/mL). Evaluation of the modulation in mRNA expression of Renogrit (10, 30, and 100 μg/mL) on Vancomycin (2 mM) induced kidney injury markers, **(F)** NGAL and **(G)** MMP7. Data represented as mean ± SEM. ###, $p < 0.001$ *vs.* Untreated control group. *, $p < 0.05$, **, $p < 0.01$, and ***, $p < 0.001$ *vs.* disease control (0) group.

## Renogrit normalized Vancomycin-induced altered kidney functions in rat model of nephrotoxicity

A rat model was developed to evaluate the protective effects of Renogrit against Vancomycin-induced acute kidney toxicity. A schematic of *in vivo* experimental methodology has been depicted in Fig 3A, and discussed in the material and methods section. Serum Blood urea nitrogen (BUN) and creatinine levels are widely used indicators to assess the renal functions [28]. Vancomycin significantly induced BUN and serum creatinine levels which were reduced ($p < 0.05$) with Renogrit co-treatment in a dose-dependent manner (Fig 3B and 3D). Similarly, BUN clearance and creatinine clearance which drastically went down in Vancomycin administered rats, significant restoration was observed with Renogrit treatment (Fig 3C and 3E). In line with these results, Renogrit also restored the urinary urea nitrogen and urinary creatinine levels (Fig 3F and 3G). The elimination of Vancomycin from the body occurs mainly by glomerular filtration and therefore a reduction in this function can impair the excretion and result in the accumulation of Vancomycin in the body which could be associated with severe adverse effects. We found in our study that estimated glomerular filtration rate is significantly decreased ($p < 0.05$) by Vancomycin and co-treatment with Renogrit reversed this reduction in eGFR in a dose dependent manner (Fig 3H) with maximal reversal observed at the dose of 600 mpk/day ($p < 0.05$).

## Renogrit reversed Vancomycin-induced oxidative stress and kidney injury markers in experimental animals

Renogrit co-treated animals showed a significant ($p < 0.01$) enhancement in their GSH/GSSG ratio (a marker of oxidative stress) which got declined ($p < 0.001$) in Vancomycin-induced animals (Fig 4A). A significant ($p < 0.001$) increase in the relative kidney weight percentage with respect to whole body was observed in Vancomycin-administrated rats. Renogrit co-treated rats showed a significant abrogation in the relative kidney weight percentage in a concentration dependent manner (Fig 4B). Besides, urine samples were analyzed for KIM-1 and NAG which are the potential biomarkers to evaluate the kidney damage. A reduction in the Vancomycin-induced KIM-1 and NAG was observed in Renogrit co-treated rat groups (Fig 4C and 4D). A significant ($p < 0.05$) reduction was also observed in the mRNA expression levels of KIM-1 and Osteopontin in Renogrit treated animals (Fig 4E and 4F).

## Renogrit ameliorated Vancomycin-induced histopathological changes in the kidney

A histopathological examination of kidney tissues revealed no pathological features such as tubular necrosis, tubular dilatation, tubular cast deposition as well as inflammatory cell infiltration, in the normal control group (Fig 5A, a). In contrast, the disease control group

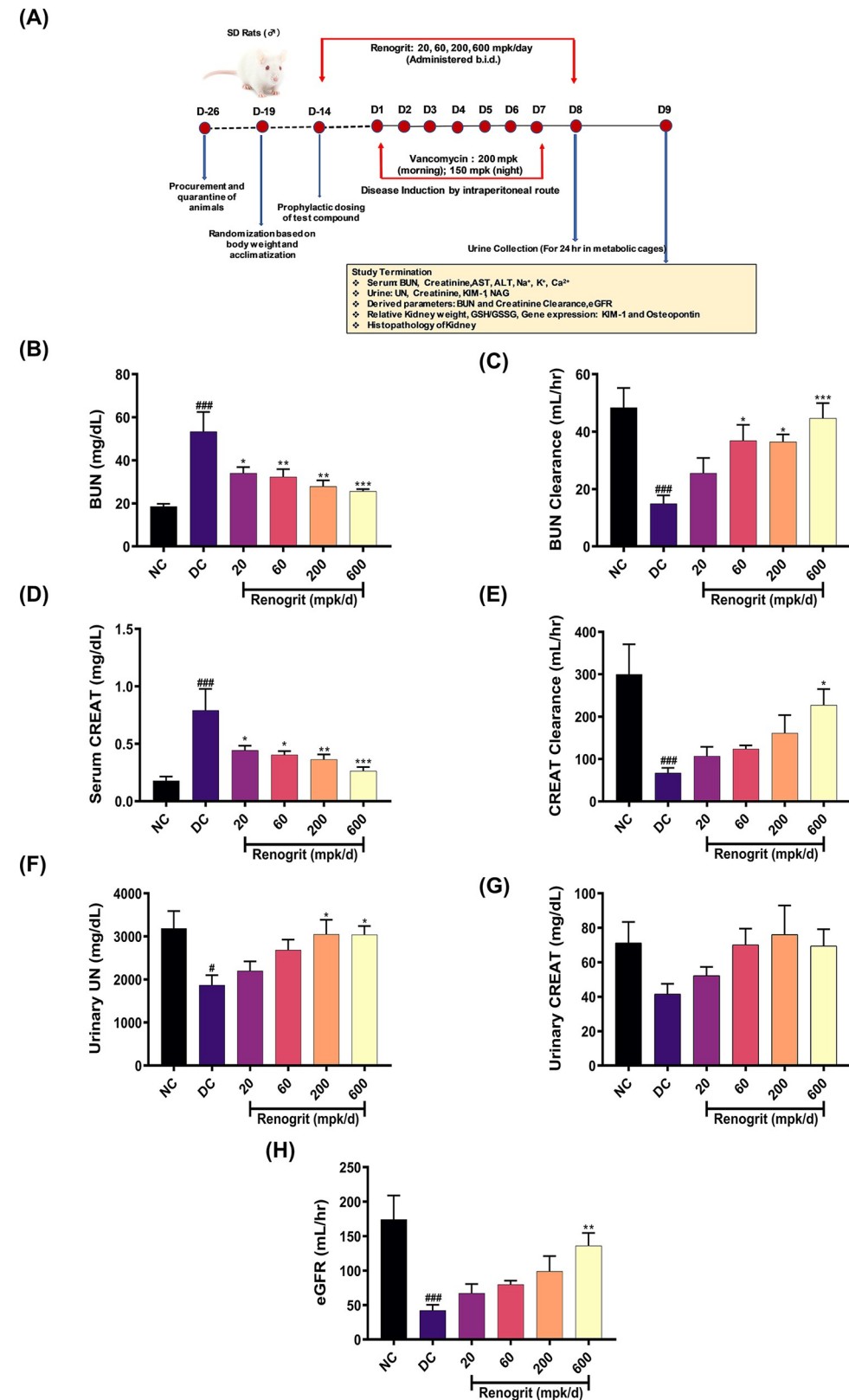

**Fig 3. *In vivo* evaluation of Renogrit in rat model of Vancomycin-induced acute kidney injury by assessment of kidney function tests. (A)** Schematic of *in vivo* experimental methodology. The alterations in **(B)** Blood Urea

Nitrogen (mg/dL) and its **(C)** clearance (mL/hr), **(D)** Serum creatinine (mg/dL) and its **(E)** clearance (mL/hr) by Vancomycin induction in rats were normalized by Renogrit (20–600 mpk/day) treatment. Similarly, normalization of **(F)** Urinary Urea Nitrogen (mg/dL), **(G)** Urinary creatinine (mg/dL) and **(H)** estimated Glomerular Filtration Rate (mL/hr) was observed in Renogrit treated groups. Data represented as mean ± SEM (n = 6). #, $p < 0.05$ and ###, $p < 0.001$ *vs.* normal control group. *, $p < 0.05$, **, $p < 0.01$, and ***, $p < 0.001$ *vs.* disease control group.

exhibited severe, diffuse tubular necrosis, tubular dilatation, deposition of casts and inflammatory cell infiltration (Fig 5A, b). Renogrit administered by oral route exhibited reduction in the tubular necrosis score with significant ($p < 0.01$) decrease at 600 mpk/day (Fig 5B). Further, all the tested doses significantly ameliorated tubular dilatation (Fig 5C) as well as tubular cast deposition at all the evaluated doses (Fig 5D). Furthermore, Renogrit also decreased Vancomycin-induced interstitial inflammatory cell infiltration (Fig 5E). Finally, the Vancomycin-induced semi-quantitative total lesion score was significantly ($p < 0.001$) reduced by all the tested doses of Renogrit (Fig 5F).

## Renogrit normalized Vancomycin-induced hematologic alterations

Intraperitoneal administration of Vancomycin in animals allocated to the disease control group for seven consecutive days, resulted in a significant increase in the total leukocyte counts (TLC) and differential leukocyte counts (DLC) in the blood, when compared to the normal control group (Table 2). Renogrit administered by oral route at the doses of 20, 60, 200 and 600 mpk/day protected the rats from the observed increase in TLC and DLC in a dose-related manner (Table 2). The observed effect for TLC was significant at Renogrit-600 mpk/day ($p < 0.01$). For absolute neutrophil count, the effect was statistically significant at all the tested doses ($p < 0.001$). Vancomycin-induced increase in absolute lymphocyte counts were significantly inhibited by Renogrit at the doses of 60 mpk/day ($p < 0.05$) and 600 mpk/day ($p < 0.01$). Further, Vancomycin-induced monocytosis was prevented by Renogrit significantly at all the evaluated doses ($p < 0.01$). For Vancomycin-induced increase in eosinophil counts, Renogrit exerted a noticeable inhibitory effect at the doses of 60 and 600 mpk/day respectively. Furthermore, Vancomycin-induced basophilia was significantly inhibited by Renogrit at the doses of 60, 200 and 600 mpk/day (Table 2). Vancomycin did not significantly alter any other evaluated hematological parameters and neither did Renogrit have any effect on RBC-related indices or the platelet counts (Table 2).

## Renogrit did not affect liver enzymes and serum electrolytes levels

In present study design, rats received Renogrit for a total of 22 consecutive days. On study termination, the levels of AST, ALT and serum electrolytes such as $Na^+$, $K^+$ and $Ca^{2+}$ were found to be comparable with the normal-control group, at all dosed tested. These findings indicated that Renogrit is broadly biocompatible and safe up to a dose of 600 mpk/day with respect to liver enzymes and serum electrolytes (Fig 6A–6E), for the duration of the study. Vancomycin administration per se also did not have any effects on the AST and levels of serum electrolytes. Interestingly, Vancomycin treatment found to reduce the ALT values ($p < 0.05$) in the disease-control group, when compared to the normal-control group (Fig 6B).

## Discussion

Vancomycin, a highly hydrophilic glycopeptide antibiotic, is a gold standard for the treatment of MRSA [10]. Its optimal bactericidal effect and low price makes it a frequently prescribed antibiotic to the hospitalized patients with infections. A major adverse effect of Vancomycin is nephrotoxicity. The Vancomycin-induced nephrotoxicity was firstly identified in 1958, after

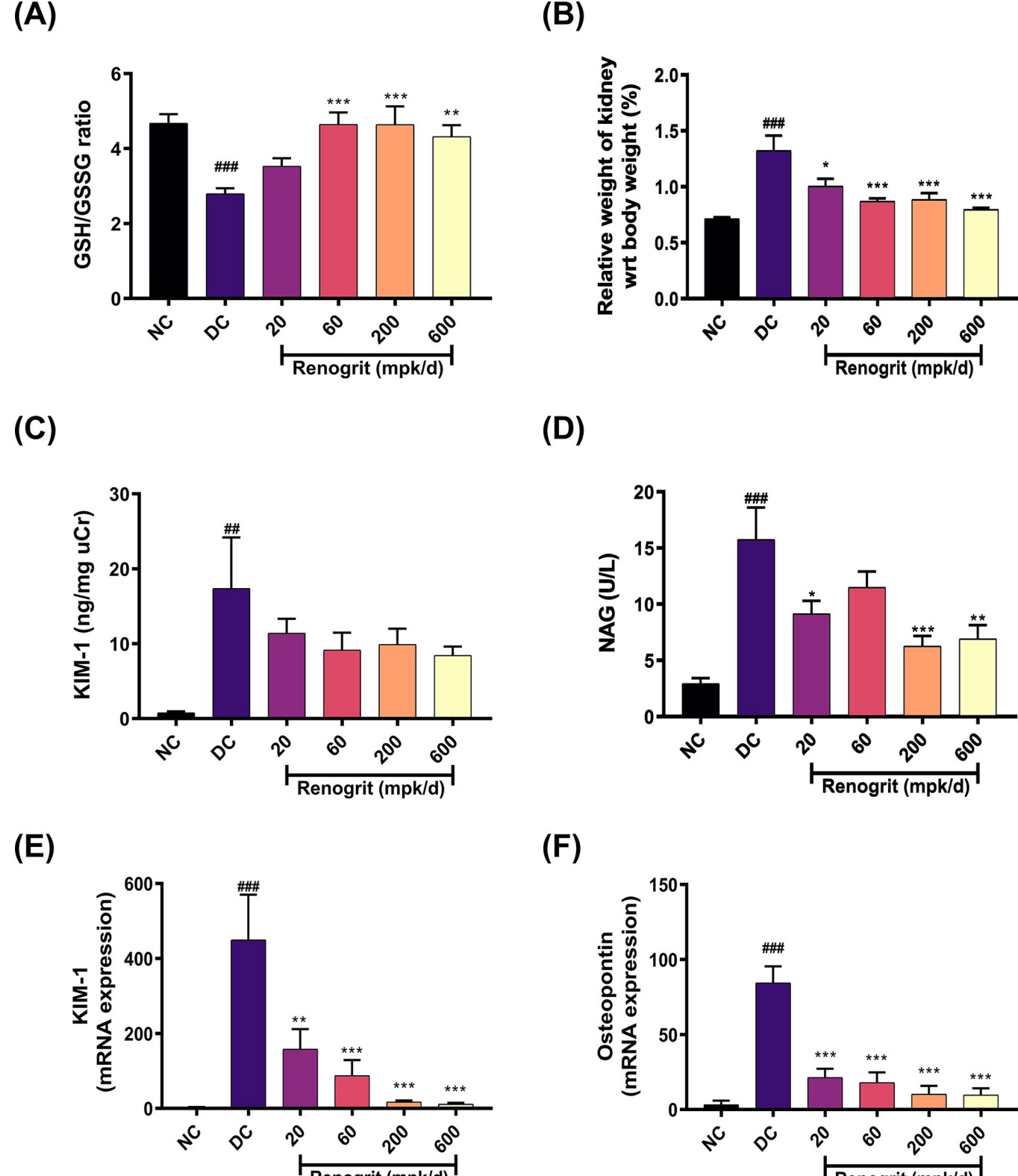

**Fig 4. Effect of Renogrit on Vancomycin-induced oxidative stress and kidney injury. (A)** Renogrit normalized the GSH/GSSG ratio in the which was altered by Vancomycin-induced oxidative stress in kidney of treated rats. **(B)** Bilateral kidneys were harvested from the sacrificed animals and weighed as detailed in the materials and methods section and the Relative kidney weights as a percentage of terminal body weight were determined. Renogrit (20–600 mpk/day)

treated rats showed a decrease in the relative kidney weights. Similarly, Renogrit treatment decreased the Vancomycin stimulated release of urinary nephrotoxicity biomarkers, **(C)** KIM-1 (ng/mg uCr) and **(D)** NAG (U/L). Also, the mRNA expression of kidney injury markers **(E)** KIM-1 and **(F)** Osteopontin was also normalized in Renogrit treated rats. Data represented as mean ± SEM (n = 6). ##, $p < 0.01$ and ###, $p < 0.001$ *vs*. normal control group. *, $p < 0.05$, **, $p < 0.01$, and ***, $p < 0.001$ *vs*. disease control group.

which several trials citing the associated nephrotoxicity have been reported. The symptoms of nephrotoxicity often emerge within 4–17 days of therapy initiation and in some cases a full recovery of renal functions does not occur. Vancomycin-induced kidney injury leads to an increase in duration of hospitalization and in some cases, mortality has also been reported [29]. The amelioration of nephrotoxicity related to Vancomycin would further augment its clinical utility. In recent times, requirement of safe therapeutic agents derived from natural resources has been increased to combat different xenobiotic induced toxicities [11].

The current study characterized the phytochemicals and pharmacological effects of Renogrit, an often-prescribed herbal medicine for kidney related ailments, against *in vitro* and *in vivo* models of Vancomycin-induced kidney injury. The phytometabolite analysis using UHPLC and MS-QToF revealed the presence of Gallic acid, Bergenin, Methyl gallate, Quercetin, Boeravinone B, Butrin, Monospermoside and Butin. These phytoconstituents have been well described for their reported anti-oxidant and anti-inflammatory activities, in the several pharmacological studies [30–38].

The *in vitro* monolayer cell culture models have been used for screening of agents which can ameliorate nephrotoxicity but these models poorly translate the normal pathologic conditions due to their homogeneity. Here, we have developed a 3D *in vitro* spheroid model using renal proximal tubule epithelial cells, HK-2. A spheroid model was developed for the evaluation of Renogrit against Vancomycin-induced tubular injury after selection of an optimum cell number and duration of treatment. This allowed our *in vitro* model to be heterogeneous and more relevant to microenvironmental pathobiology of kidney tubular damage [6].

Primarily, we assessed the cytosafety of Renogrit before assessing its pharmacological activity in HK-2 spheroids. It was found to be safe at all the tested concentrations (10–100 μg/mL). Further evaluation of Renogrit was carried at 2 mM concentration of Vancomycin which was sufficient for inducing various parameters associated with the kidney damage. Initially, we tested Renogrit against Vancomycin induced cytotoxicity in HK-2 spheroids and found it significantly reversed the decrease in viability in a dose dependent manner. These cytoprotective effects of Renogrit can be linked to the presence of Gallic acid, one of the phytoconstituents of Renogrit [39]. P-glycoprotein (P-gp) is an ATP-dependent efflux protein that is present in a variety of normal tissues, and it is abundantly expressed at the apical membrane of the kidney proximal tubules. The P-gp plays an important role in the efflux of variety of chemicals including drugs into urine [2,16,40]. We detected that Vancomycin decreased the efflux capacity of HK-2 spheroids as depicted from fluorescence dye accumulation, but co-treatment with Renogrit significantly attenuated this increase. Furthermore, it was observed that decrease in dye accumulation with Renogrit is comparable with that from Cilastatin which is reported to exert the protective effect in HK-2 cells, as well as in the rat model of Vancomycin induced nephrotoxicity [16]. This effect of Renogrit on P-gp could well be due to the presence of Quercetin which is a known P-gp inducer [41].

Kidney injury biomarkers like NAG, KIM-1, NGAL, and MMP-7 were also assessed on the Vancomycin-induced HK2 spheroids. These markers are expressed in the low levels but get significantly upregulated following kidney injury [15,24–26]. The release and expression of these biomarkers declined significantly with Renogrit co-treatment in a dose-dependent manner. These protective effects of Renogrit can be due to the presence of Bergenine and Butin as its major phytoconstituents which possesses anti-oxidant properties.

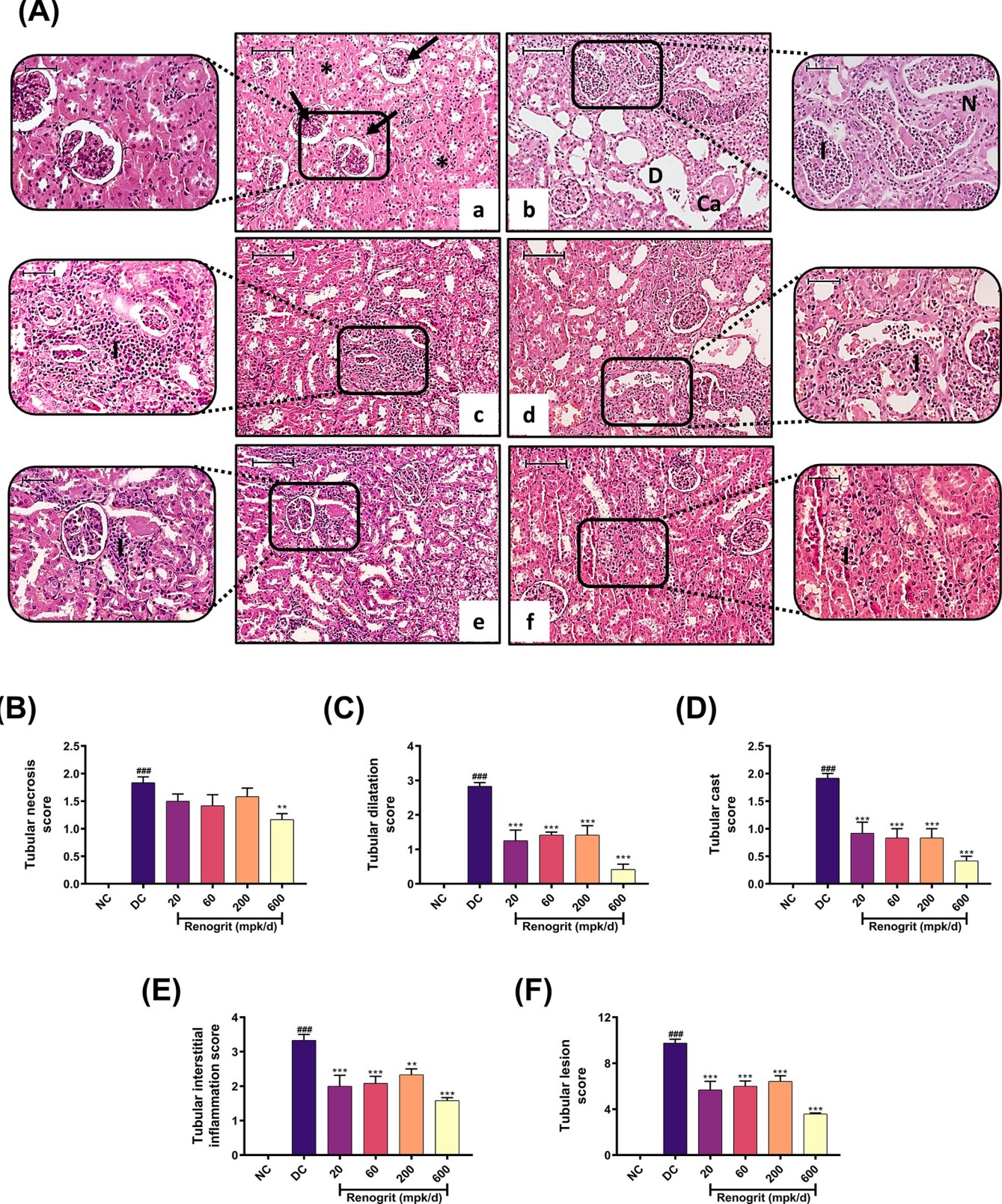

**Fig 5. Effect of Renogrit on Vancomycin-induced altered histoarchitecture of the kidneys.** The neutral buffered formalin-fixed left kidney was subjected to standard tissue processing procedures and tissue blocks were sectioned. The obtained sections were stained with hematoxylin and eosin as mentioned in the

materials and method section. **(A)** Representative photomicrographic images (10× and 20×, Scale = 100 μm) of the kidney of (a) Normal control group rat showing normal renal architecture consisting of glomeruli (arrow) and tubules (star) uniformly organized throughout the section; (b) DC group rat exhibiting severe tubular necrosis appearing as loss of normal architecture (N) along with inflammatory cell infiltration (I), tubular dilation (D) and deposition of tubular cast (Ca); (c-f) Renogrit treated rats exhibiting ameliorating effect on the altered kidney structure and overall restoration of normal renal architecture at the highest tested dose. **(B)** Tubular necrosis score, **(C)** Tubular dilatation score, **(D)** Tubular cast deposition score, **(E)** Tubular interstitial inflammation score, and **(F)** Total lesion score. Data represented as mean ± SEM (n = 6). ###, $p < 0.001$ vs. normal control group. $p < 0.01$ and ***, $p < 0.001$ vs. disease control group.

After we established pharmacological effects of Renogrit *in vitro*, we further evaluated its potential protective effects at different doses (20, 60, 200 and 600 mpk/day) in the rat model of Vancomycin-induced nephrotoxicity. Significantly increased serum BUN and creatinine levels are associated with compromised renal functions in response to Vancomycin insult [42]. In the present study also, rats injected with Vancomycin showed increased levels of BUN and creatinine but those co-treated with Renogrit showed normalized levels of BUN and creatinine. A modulation in the kidney functions causes an impairment in the urinary clearance of waste products from the blood which leads to an increase in the blood urea and creatinine level but Renogrit treated group showed a normal clearance rate of BUN and creatinine. Also, eGFR values decline due to Vancomycin-induced injury [43]. Interestingly, the Renogrit co-treated groups showed a dose-dependent recovery in their eGFR. This can be linked to the decreased renal injury due to the antioxidant effects of Renogrit.

The injury caused by Vancomycin is majorly due to the development of oxidative stress [3,10,20,44]. Glutathione (GSH) is the most abundant antioxidant in aerobic cells, is critical for protection from oxidative stress, acting as a free radical scavenger and inhibitor of lipid peroxidation. GSH also participates in the detoxification of hydrogen peroxide by various glutathione peroxidases. The ratio of reduced GSH to oxidized GSH (GSSG) is an indicator of

**Table 2. Effect of Renogrit on hematological parameters.**

| Parameter | Normal control | Disease control | Renogrit (20 mpk/d) | Renogrit (60 mpk/d) | Renogrit (200 mpk/d) | Renogrit (600 mpk/d) |
|---|---|---|---|---|---|---|
| TLC ($10^3$/μL) | 10.65 ± 0.86 | 19.37 ± 1.80## | 17.17 ± 1.61 | 13.6 ± 1.64 | 14.55 ± 2.17 | 11.4 ± 1.59** |
| NEU ($10^3$/μL) | 1.52 ± 0.07 | 10.26 ± 0.79### | 6.47 ± 0.79*** | 3.15 ± 0.26*** | 4.57 ± 0.54*** | 2.66 ± 0.23*** |
| LYM ($10^3$/μL) | 6.01 ± 0.07 | 10.09 ± 1.03### | 9.33 ± 0.72 | 7.52 ± 0.25* | 8.28 ± 0.54 | 6.96 ± 0.23** |
| MONO ($10^3$/μl) | 0.31 ± 0.03 | 1.41 ± 0.19### | 0.79 ± 0.13** | 0.51 ± 0.04*** | 0.72 ± 0.06*** | 0.45 ± 0.07*** |
| EOS ($10^3$/μL) | 0.08 ± 0.01 | 0.26 ± 0.04## | 0.24 ± 0.05 | 0.16 ± 0.01 | 0.22 ± 0.03 | 0.16 ± 0.03 |
| BASO ($10^3$/μL) | 0.05 ± 0.01 | 0.24 ± 0.02### | 0.17 ± 0.03 | 0.12 ± 0.02** | 0.14 ± 0.03* | 0.08 ± 0.01*** |
| RBC ($10^6$/μL) | 7.30 ± 0.19 | 7.32 ± 0.48 | 6.80 ± 0.37 | 6.62 ± 0.67 | 6.86 ± 0.5837 | 7.39 ± 0.29 |
| HGB (g/dL) | 14.90 ± 0.45 | 14.13 ± 0.79 | 13.28 ± 0.70 | 13.07 ± 1.10 | 13.62 ± 0.99 | 14.15 ± 0.47 |
| HCT (%) | 39.92 ± 1.09 | 38.12 ± 2.12 | 35.80 ± 1.91 | 35.48 ± 2.70 | 36.35 ± 2.43 | 38.03 ± 1.51 |
| MCV (fL) | 54.72 ± 0.59 | 52.28 ± 0.72 | 52.73 ± 0.49 | 54.47 ± 1.83 | 53.77 ± 2.07 | 51.53 ± 0.57 |
| MCH (pg) | 20.38 ± 0.25 | 19.37 ± 0.23 | 19.55 ± 0.17 | 19.92 ± 0.47 | 20.07 ± 0.60 | 19.18 ± 0.29 |
| MCHC (g/dL) | 37.30 ± 0.22 | 37.03 ± 0.08 | 37.08 ± 0.23 | 36.65 ± 0.45 | 37.38 ± 0.31 | 37.22 ± 0.28 |
| PLT ($10^3$/μL) | 1160.00 ± 75.44 | 1177.00 ± 207.80 | 1240.00 ± 154.20 | 1441.00 ± 193.20 | 1348.00 ± 65.89 | 1109.00 ± 90.36 |

TLC, total leukocyte count; NEU, neutrophil count, LYM, lymphocyte count; MONO, monocyte count; EOS, eosinophil count; BASO, basophil count; RBC, red blood cell count; HGB, Hemoglobin; HCT, hematocrit; MCV, mean corpuscular volume; MCH, mean corpuscular hemoglobin; MCHC, mean corpuscular hemoglobin concentration; PLT, platelet count. Data presented as mean ± SEM (n = 6).

## $p < 0.01$

### $p < 0.001$ *vs.* normal control

* $p < 0.05$

** $p < 0.01$ and

*** $p < 0.001$ *vs.* disease control.

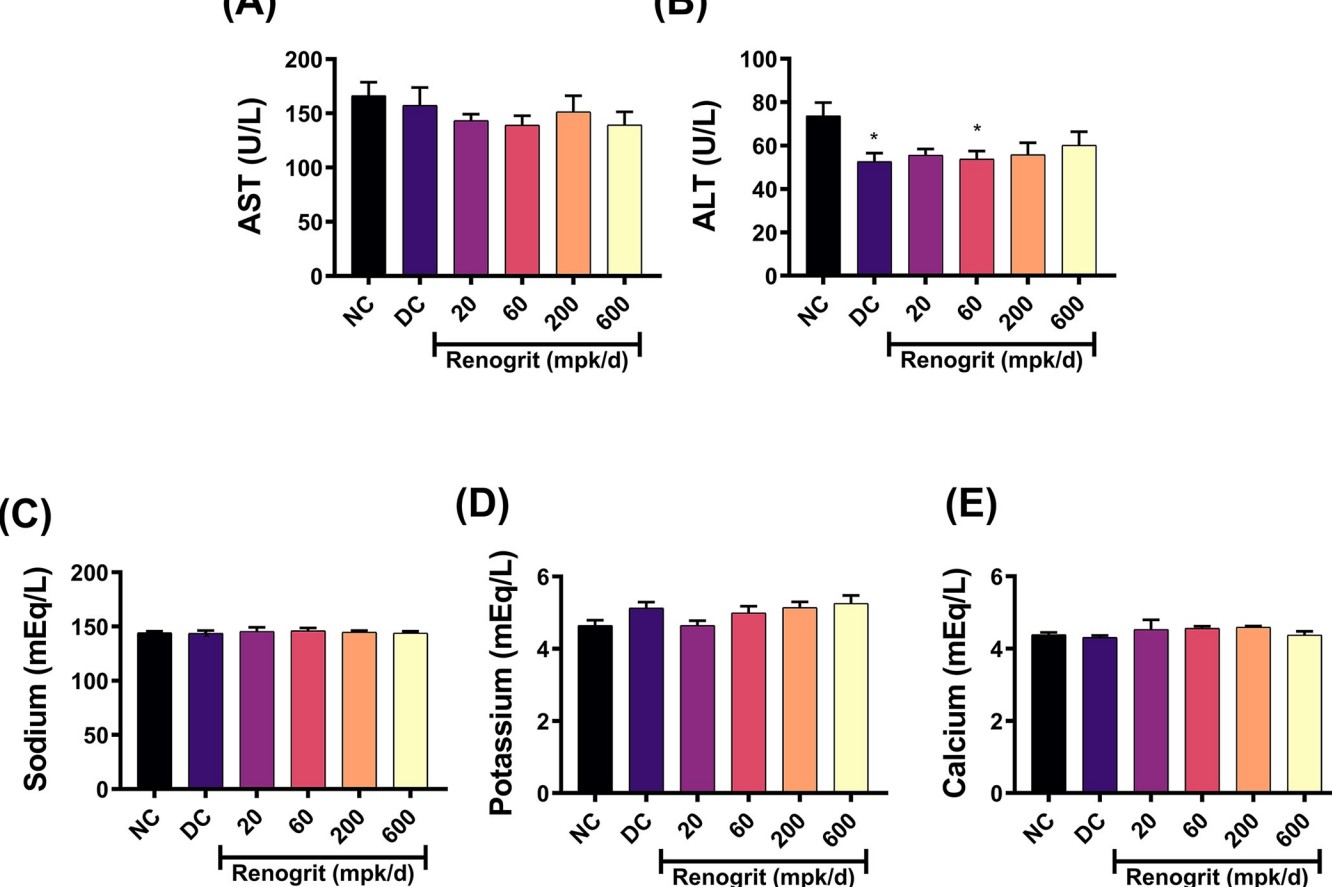

**Fig 6. Effect of Renogrit on liver function and serum electrolytes.** Serum was subjected to clinical chemistry analysis to estimate the levels of **(A)** AST, **(B)** ALT, **(C)** $Na^+$, **(D)** $K^+$, and **(E)** $Ca^{2+}$ as elaborated in materials and methods section. Data represented as mean ± SEM (n = 6). *, $p < 0.05$ *vs.* normal control group.

cellular health and an excellent way to assess potential therapeutics' efficacy in maintaining cellular redox potential [45]. In our study we observed that Renogrit treatment normalized the Vancomycin induced reduction in GSH/GSSG ratio. Therefore, Renogrit treatment can halt the subsequent damages to renal tubules caused by Vancomycin induced oxidative stress. This might be due to the presence of methyl gallate a potent antioxidant [46] in Renogrit.

An increase in the weight of the kidney in response to Vancomycin insult was observed which got normalized in a dose-dependent fashion in Renogrit co-treated animals. Such an increase in kidney weight post Vancomycin treatment was also reported by Yu *et al.* [3] wherein they observed that antioxidants were able to decrease such pathological event. The increase in kidney weight was accompanied by an increase in the kidney injury markers namely KIM-1, NAG and osteopontin [5,47]. However, in the rats treated with Renogrit the insult to kidney in response to Vancomycin treatment was curbed as a result of which an apparent decrease in the levels of kidney injury biomarkers was observed. This effect of Renogrit is due to the presence of Boeravinone B which is a known nephroprotective agent [48].

The renal histopathological damages associated with Vancomycin treatment are assessed by scoring the observed tubular necrosis, dilatation, cast formation, interstitial edema, and lesions [3,19]. In our present study we found that these histopathological changes were ameliorated in Renogrit treated groups. Interestingly, a significant decrease in the tubular cast formation was

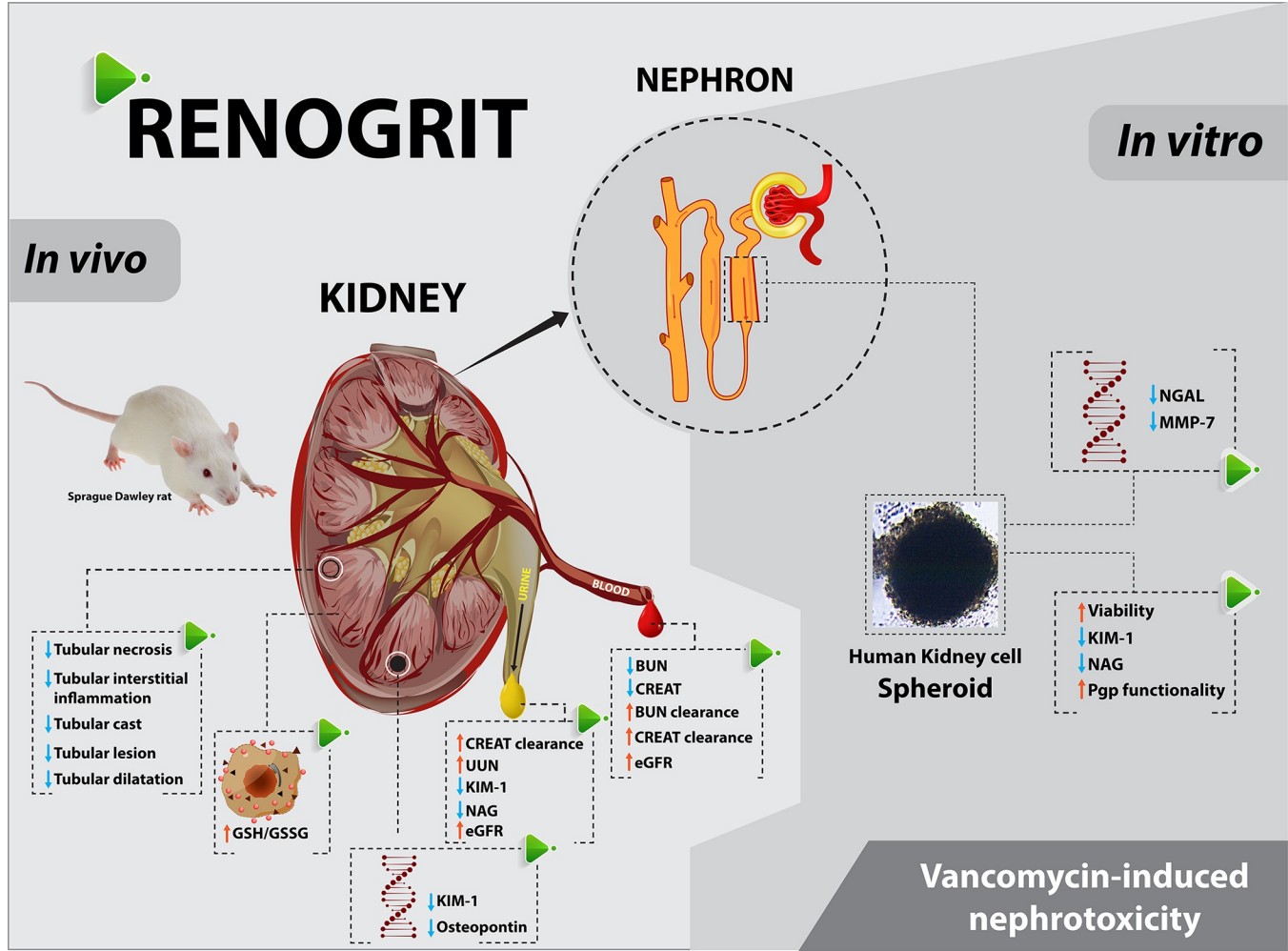

**Fig 7. Summary of the biochemical, histological and molecular changes induced by Vancomycin in HK-2 spheroids and SD rats along-with subsequent therapeutic modulation by Renogrit.**

also observed in Renogrit treated animals. The Vancomycin-associated tubular casts (VTC) are known to get localized in the tubules, block the urine flow, trigger localized necrosis of tubular epithelial cells, inflammation and consequently cause kidney injury. An increasing body of evidence suggests that VTC are the main cause of the nephrotoxic effects of Vancomycin induced kidney injury [10,29,49,50]. Uromodulin, the most abundant urinary protein produced by renal epithelial cells interacts with Vancomycin aggregates and forms an intratubular cast [10]. But uromodulin is not only involved in cast formation, it also acts as a danger-associated–molecular pattern molecule (DAMP) and recruit macrophages at the site of tissue injury leading to sterile inflammation [51–53]. The increase in the levels of leukocytes, neutrophils, monocytes, eosinophils, and basophils also direct towards the increase in sterile inflammation due to Vancomycin-uromodulin aggregate mediated tubular necrosis [54–57]. This increase in the hematology parameters was subdued in Renogrit treated groups which can be related to the decrease observed in the tubular necrosis and inflammation score. The anti-inflammatory effects of Renogrit might be in part due to presence of the anti-inflammatory phytochemical Butrin [58]. Taken together, Renogrit treatment decreased the tubular injury and subsequent inflammation caused by Vancomycin. Thus, Renogrit has treatment led to a reduction in

oxidative stress induced by vancomycin. It also prevented the downstream damages namely inflammation and cell injury as observed from the histopathological and biochemical assessments. A summary of the pharmacological effects of Renogrit against Vancomycin-induced nephrotoxicity has been mentioned in Fig 7.

In order to rule out any confounding bias in our results due to any effect of Vancomycin on liver we also performed the analysis of serum ALT, AST and electrolytes namely $Na^+$, $K^+$, $Ca^{2+}$. In our study we did not find any major alteration in the levels of serum aminotransferases and electrolytes. Hence, neither Vancomycin nor Renogrit produced any ill-effect on liver. Therefore, all the kidney function test parameters that are closely linked with the liver [59] were not affected by any changes related to liver.

## Conclusion

The 3D-*in vitro* and *in vivo* assessment of Renogrit against Vancomycin induced nephrotoxicity asserts the effectiveness of Renogrit. The decrease in the toxicity of Vancomycin-induced HK2 spheroids, normalization of their P-gp functionality and a decline of tubular injury markers was observed in Renogrit co-treated spheroids. Moreover, the apparent amelioration of kidney injury markers, normalization of creatinine and urea clearance, histopathological findings in response to Renogrit treatment point towards the clinico-therapeutic potential of the antioxidant phytomedicine-Renogrit in alleviation of kidney toxicity frequently encountered with the use of Vancomycin.

## Supporting information

**S1 Fig. Viability assessment.**
(DOCX)

## Acknowledgments

We extend our gratitude to Ms. Meenu Tomer, Mr. Sudeep Verma, Dr. Seema Gujral and Dr. Jyotish Srivastava for their support in the phytochemical analysis. We are thankful Ms. Moumita Manik for her support in spheroid preparation. We are thankful to Dr. Rani Singh for her support in gene expression analysis. We are grateful to Dr. Tapan Dey, Ms. Deepika Kumari, Ms Deepika Mehra, Mr. Ram Hari Sharma, Mr. Pushpendra Singh, Mr. Sonit Kumar and Mr. Rajat Kumar for their support in the *in vivo* analysis. We are thankful to Mr. Devendra Kumawat for his help in preparation of summary figure. We are also grateful to Mr. Tarun Rajput and Mr. Gagan Kumar for their swift administrative support.

## Author Contributions

**Conceptualization:** Acharya Balkrishna, Vivek Gohel, Anurag Varshney.

**Data curation:** Vivek Gohel, Sandeep Sinha, Rishabh Dev.

**Formal analysis:** Sonam Sharma, Vivek Gohel, Ankita Kumari, Malini Rawat, Madhulina Maity, Sandeep Sinha.

**Investigation:** Sonam Sharma, Vivek Gohel, Ankita Kumari, Malini Rawat, Madhulina Maity, Sandeep Sinha.

**Methodology:** Sonam Sharma, Vivek Gohel, Ankita Kumari, Malini Rawat, Madhulina Maity, Sandeep Sinha.

**Project administration:** Acharya Balkrishna, Sandeep Sinha, Rishabh Dev, Anurag Varshney.

**Resources:** Acharya Balkrishna, Anurag Varshney.

**Supervision:** Acharya Balkrishna, Sandeep Sinha, Rishabh Dev, Anurag Varshney.

**Visualization:** Acharya Balkrishna, Sonam Sharma.

**Writing – original draft:** Vivek Gohel, Rishabh Dev.

**Writing – review & editing:** Acharya Balkrishna, Vivek Gohel, Rishabh Dev, Anurag Varshney.

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
