## [Decision Letter · Decision Letter 0]

17 Jul 2023

PONE-D-23-14497Renogrit attenuates Vancomycin-induced nephrotoxicity in human renal tubular spheroids and SD rats by regulating creatinine/urea clearance and kidney injury biomarkersPLOS ONE

Dear Dr. Varshney,

Thank you for submitting your manuscript to PLOS ONE. After careful consideration, we feel that it has merit but does not fully meet PLOS ONE’s publication criteria as it currently stands. Therefore, we invite you to submit a revised version of the manuscript that addresses the points raised during the review process.

 Please submit your revised manuscript by Aug 31 2023 11:59PM. If you will need more time than this to complete your revisions, please reply to this message or contact the journal office at plosone@plos.org. Please include the following items when submitting your revised manuscript:A rebuttal letter that responds to each point raised by the academic editor and reviewer(s). You should upload this letter as a separate file labeled 'Response to Reviewers'.A marked-up copy of your manuscript that highlights changes made to the original version. You should upload this as a separate file labeled 'Revised Manuscript with Track Changes'.An unmarked version of your revised paper without tracked changes. You should upload this as a separate file labeled 'Manuscript'.

We look forward to receiving your revised manuscript.

Kind regards,

Ahmed E. Abdel Moneim

Academic Editor

PLOS ONE

“The test article was provided by Divya Pharmacy, Haridwar, Uttarakhand, India. Acharya Balkrishna is an honorary trustee in Divya Yog Mandir Trust, which governs Divya Pharmacy, Haridwar. In addition, he holds an honorary managerial position in Patanjali Ayurved Ltd, Haridwar, India. Other than providing the test formulation (Renogrit), Divya Pharmacy was not involved in any aspect of the research reported in this study. All other authors have declared no conflict of interest.”

3. We note that Figure 7 in your submission contain copyrighted images. All PLOS content is published under the Creative Commons Attribution License (CC BY 4.0), which means that the manuscript, images, and Supporting Information files will be freely available online, and any third party is permitted to access, download, copy, distribute, and use these materials in any way, even commercially, with proper attribution. For more information, see our copyright guidelines: http://journals.plos.org/plosone/s/licenses-and-copyright.

a. You may seek permission from the original copyright holder of Figure 7  to publish the content specifically under the CC BY 4.0 license.

Reviewers' comments:

Reviewer's Responses to Questions

**Comments to the Author**

1. Is the manuscript technically sound, and do the data support the conclusions?

Reviewer #1: Yes

Reviewer #2: Yes

Reviewer #3: Yes

2. Has the statistical analysis been performed appropriately and rigorously? 

Reviewer #1: Yes

Reviewer #2: Yes

Reviewer #3: Yes

3. Have the authors made all data underlying the findings in their manuscript fully available?

Reviewer #1: Yes

Reviewer #2: Yes

Reviewer #3: Yes

4. Is the manuscript presented in an intelligible fashion and written in standard English?

Reviewer #1: No

Reviewer #2: Yes

Reviewer #3: Yes

5. Review Comments to the Author

Reviewer #1: In the current work, the authors investigated the protective effects of Renogrit against Vancomycin-induced nephrotoxicity both in vitro and in vivo. This a rich interesting study. However, I have some suggestions to improve the manuscript.

1. It would be better to use the full term "Sprague Dawley" in the title rather than the abbreviation "SD".

2. What is the basis of selection of the treatment doses in the in vivo study? The authors explained how they calculated the therapeutic dose, but gave no clue of the selection of other doses. For example, why was a dose 3 times the therapeutic dose used? Why not 2 times? Why 1/3 the therapeutic dose? Why not 1/2???!!!!

3. What is the reference of the recommended human dose of Renogrit which the authors used to calculate the therapeutic dose in rats?

4. A reference of the dose and duration of Vancomycin used to induce kidney injury is required.

5. In figures 3 and 4, the use of significance symbols was inaccurate. The authors ignored to add them many times. The use of significance symbols should be revised accurately on all the charts' columns.

6. The description of the results in table 1 should be revised to cope with the results presented in the table. For example, the authors mentioned that "For Vancomycin-induced increase in eosinophil counts, Renogrit exerted inhibitory effects at the doses of 60 and 600 mpk/day respectively", while the tables shows no significant changes in all treated groups.

7. In figure 7, UUN was estimated in the urine not in the blood.

8. The language of the manuscript should be revised thoroughly.

Reviewer #2: 1. This study is to test the protective effect and mechanism of Renogrit on kidney injury from the cell level to animal experiments. The analysis strategy of this study is step by step and the experimental design and process are very rigorous and consistent.

2. The choice of dosage is well and correctly described.

3. Manuscript writing is easy to understand and provides evidence from recent literature.

4. Lines 47 to 51 on page 3, the words are repetitive and lengthy, please modify them.

5. This manuscript is of reference value to the academic community and complies with the standard of PLOS ONE journals. It is recommended to publish.

Reviewer #3: - The Title of the article should be shortened.

- The keywords must be selected from MeSH-NCBI.

- In the introduction section, the mechanism of Vancomycin nephrotoxicity should be further explained.

- All parts of the Method must be appropriate referenced.

- How were the doses and times selected? Are they based on the literature or on epidemiologic/exposure studies?

- What statistical test was used for each experiment?

- Discussion should be reviewed and rewritten with existing literature. There is a lack of mechanism.

- It is recommended an English revision for the text. Some grammar errors should be corrected.

- Figures or Graphs should be clearer. They are not clear to the reader.

6. PLOS authors have the option to publish the peer review history of their article (what does this mean?). If published, this will include your full peer review and any attached files.

Reviewer #1: No

Reviewer #2: No

Reviewer #3: No

---

## [Author Response · Author response to Decision Letter 0]

22 Jul 2023

Academic editor

Comment: Please ensure that your manuscript meets PLOS ONE's style requirements, including those for file naming. The PLOS ONE style templates can be found at https://journals.plos.org/plosone/s/file?id=wjVg/PLOSOne_formatting_sample_main_body.pdf and https://journals.plos.org/plosone/s/file?id=ba62/PLOSOne_formatting_sample_title_authors_affiliations.pdf

Response: We thank the editor for the remark. We have gone through the journal requirements and have revised the formatting as per the suggestions.

Comment: Thank you for stating the following in the Competing Interests section: “The test article was provided by Divya Pharmacy, Haridwar, Uttarakhand, India. Acharya Balkrishna is an honorary trustee in Divya Yog Mandir Trust, which governs Divya Pharmacy, Haridwar. In addition, he holds an honorary managerial position in Patanjali Ayurved Ltd, Haridwar, India. Other than providing the test formulation (Renogrit), Divya Pharmacy was not involved in any aspect of the research reported in this study. All other authors have declared no conflict of interest.” Please confirm that this does not alter your adherence to all PLOS ONE policies on sharing data and materials, by including the following statement: "This does not alter our adherence to PLOS ONE policies on sharing data and materials.” (as detailed online in our guide for authors http://journals.plos.org/plosone/s/competing-interests). If there are restrictions on sharing of data and/or materials, please state these. Please note that we cannot proceed with consideration of your article until this information has been declared. Please include your updated Competing Interests statement in your cover letter; we will change the online submission form on your behalf.

Response: We thank the editor for the comment. Kindly consider our revised Competing interests statement:

“The test article was provided by Divya Pharmacy, Haridwar, Uttarakhand, India. Acharya Balkrishna is an honorary trustee in Divya Yog Mandir Trust, which governs Divya Pharmacy, Haridwar. In addition, he holds an honorary managerial position in Patanjali Ayurved Ltd, Haridwar, India. Other than providing the test formulation (Renogrit), Divya Pharmacy was not involved in any aspect of the research reported in this study. This does not alter our adherence to PLOS ONE policies on sharing data and materials. All other authors have declared no conflict of interest.”

We have included the statement in our cover letter as well.

Comment: We note that Figure 7 in your submission contain copyrighted images. All PLOS content is published under the Creative Commons Attribution License (CC BY 4.0), which means that the manuscript, images, and Supporting Information files will be freely available online, and any third party is permitted to access, download, copy, distribute, and use these materials in any way, even commercially, with proper attribution. For more information, see our copyright guidelines: http://journals.plos.org/plosone/s/licenses-and-copyright. We require you to either (1) present written permission from the copyright holder to publish these figures specifically under the CC BY 4.0 license, or (2) remove the figures from your submission: a. You may seek permission from the original copyright holder of Figure 7 to publish the content specifically under the CC BY 4.0 license. We recommend that you contact the original copyright holder with the Content Permission Form (http://journals.plos.org/plosone/s/file?id=7c09/content-permission-form.pdf) and the following text: “I request permission for the open-access journal PLOS ONE to publish XXX under the Creative Commons Attribution License (CCAL) CC BY 4.0 (http://creativecommons.org/licenses/by/4.0/). Please be aware that this license allows unrestricted use and distribution, even commercially, by third parties. Please reply and provide explicit written permission to publish XXX under a CC BY license and complete the attached form.” Please upload the completed Content Permission Form or other proof of granted permissions as an "Other" file with your submission. In the figure caption of the copyrighted figure, please include the following text: “Reprinted from [ref] under a CC BY license, with permission from [name of publisher], original copyright [original copyright year].” b. If you are unable to obtain permission from the original copyright holder to publish these figures under the CC BY 4.0 license or if the copyright holder’s requirements are incompatible with the CC BY 4.0 license, please either i) remove the figure or ii) supply a replacement figure that complies with the CC BY 4.0 license. Please check copyright information on all replacement figures and update the figure caption with source information. If applicable, please specify in the figure caption text when a figure is similar but not identical to the original image and is therefore for illustrative purposes only.

Response: We are grateful to the editor for the comment. The images in Figure 7 were downloaded from Shutterstock (NY, USA) for which we have a subscription (User ID: 317535365). The images were than modified as per our observations from the study. As the images were part of our paid subscription, they will not come under the copyright protection. Except for the image of HK-2 cell spheroid which was clicked from our microscope as part of this study for which copyright does not apply.

Comment: In your Data Availability statement, you have not specified where the minimal data set underlying the results described in your manuscript can be found. PLOS defines a study's minimal data set as the underlying data used to reach the conclusions drawn in the manuscript and any additional data required to replicate the reported study findings in their entirety. All PLOS journals require that the minimal data set be made fully available. For more information about our data policy, please see http://journals.plos.org/plosone/s/data-availability. Upon re-submitting your revised manuscript, please upload your study’s minimal underlying data set as either Supporting Information files or to a stable, public repository and include the relevant URLs, DOIs, or accession numbers within your revised cover letter. For a list of acceptable repositories, please see http://journals.plos.org/plosone/s/data-availability#loc-recommended-repositories. Any potentially identifying patient information must be fully anonymized. Important: If there are ethical or legal restrictions to sharing your data publicly, please explain these restrictions in detail. Please see our guidelines for more information on what we consider unacceptable restrictions to publicly sharing data: http://journals.plos.org/plosone/s/data-availability#loc-unacceptable-data-access-restrictions. Note that it is not acceptable for the authors to be the sole named individuals responsible for ensuring data access. We will update your Data Availability statement to reflect the information you provide in your cover letter.

Response: We are thankful to the editor for the comment. All the data generated to get to the conclusion of the study is provided in the manuscript as well as the supplementary information. Kindly consider our revised Data Availability statement:

“The data supporting the findings of this study are available within the article and its supplementary material.”

We have included the statement in our cover letter as well.

Reviewer 1

In the current work, the authors investigated the protective effects of Renogrit against Vancomycin-induced nephrotoxicity both in vitro and in vivo. This a rich interesting study. However, I have some suggestions to improve the manuscript.

Comment: It would be better to use the full term "Sprague Dawley" in the title rather than the abbreviation "SD".

Response: We thank the reviewer for the suggestion. The full term "Sprague Dawley" has been added in the title instead of SD.

Comment: What is the basis of selection of the treatment doses in the in vivo study? The authors explained how they calculated the therapeutic dose, but gave no clue of the selection of other doses. For example, why was a dose 3 times the therapeutic dose used? Why not 2 times? Why 1/3 the therapeutic dose? Why not 1/2???!!!!

Response: We are grateful for the query. The dose selection was based on half-log decremental and incremental values of the therapeutic doses. This was done to space out the treatment doses and to have clearer pharmacological effects. In our experience, we have seen if the doses are close to each other, sometimes the pharmacological response is rather indistinguishable.

Comment: What is the reference of the recommended human dose of Renogrit which the authors used to calculate the therapeutic dose in rats?

Response: We thank the reviewer for the remark. The recommended human dose has been described by the manufacturer (Divya Pharmacy, Haridwar) for optimal medicinal benefits, which is based on ancient Ayurvedic text Bhavprakash Nighantu. We have followed the same recommendation.

Comment: A reference of the dose and duration of Vancomycin used to induce kidney injury is required.

Response: We acknowledge the query of reviewer. The relevant references have now been added in the methods section.

Comment: In figures 3 and 4, the use of significance symbols was inaccurate. The authors ignored to add them many times. The use of significance symbols should be revised accurately on all the charts' columns.

Response: We humbly thank the esteemed reviewer for this excellent observation. The use of significance symbol has now been corrected in Figure 3 and 4 as per the p-values obtained post data analysis by ANOVA. In the graphs wherein the significance symbols are not mentioned, the p-values were found to be less than 0.05, however the rise or decline in the biomarkers was visually noticeable. 

Comment: The description of the results in table 1 should be revised to cope with the results presented in the table. For example, the authors mentioned that "For Vancomycin-induced increase in eosinophil counts, Renogrit exerted inhibitory effects at the doses of 60 and 600 mpk/day respectively", while the tables shows no significant changes in all treated groups.

Response: We are grateful to the reviewer for the comments. The description has now been modified as per the data in the table (Page no. 23; line no. 496-497).

Comment: In figure 7, UUN was estimated in the urine not in the blood.

Response: We are grateful to the reviewer for the observation. The figure 7 has now been revised.

Comment: The language of the manuscript should be revised thoroughly.

Response: We acknowledge the comment by the reviewer. The manuscript has now been thoroughly revised linguistically.

Reviewer 2

Comment: This study is to test the protective effect and mechanism of Renogrit on kidney injury from the cell level to animal experiments. The analysis strategy of this study is step by step and the experimental design and process are very rigorous and consistent.

Response: We thank the reviewer for the remarks.

Comment: The choice of dosage is well and correctly described.

Response: We acknowledge the comment by the reviewer.

Comment: Manuscript writing is easy to understand and provides evidence from recent literature.

Response: We are grateful for the comment of the reviewer.

Comment: Lines 47 to 51 on page 3, the words are repetitive and lengthy, please modify them.

Response: We thank the reviewer for the comment. As per the suggestion, the repetitive lines have now been removed from the introduction section.

Comment: This manuscript is of reference value to the academic community and complies with the standard of PLOS ONE journals. It is recommended to publish.

Response: We highly appreciate the recommendation of the reviewer and thank him for his remarks.

Reviewer 3

Comment: The Title of the article should be shortened.

Response: We thank the reviewer for the remark. The revised title now reads as: Renogrit Attenuates Vancomycin-induced Nephrotoxicity in Human Renal Spheroids and in Sprague-Dawley Rats by Regulating Kidney Injury Biomarkers and Creatinine/Urea Clearance

Comment: The keywords must be selected from MeSH-NCBI.

Response: We acknowledge the remark from the reviewer. The keywords have now been added in the manuscript as per MeSH-NCBI.

Comment: In the introduction section, the mechanism of Vancomycin nephrotoxicity should be further explained.

Response: We acknowledge this comment by the reviewer. The mechanism of Vancomycin induced nephrotoxicity has now been added in the introduction section (Page no. 3; line no. 50-53).

Comment: All parts of the Method must be appropriate referenced.

Response: We thank the reviewer for this advice. Additional references have been added in the methods section.

Comment: How were the doses and times selected? Are they based on the literature or on epidemiologic/exposure studies?

Response: We are grateful to the reviewer for the remark. The doses and time selected were as per the published literature for vancomycin; and as per manufacturer recommendations for Renogrit. The references have been added in method section of the manuscript.

Comment: What statistical test was used for each experiment?

Response: We thank the reviewer for the query. A one-way analysis of variance (ANOVA), which was followed by Dunnett’s multiple comparison post-hoc test was employed to compute the statistical differences between the mean values. A p value < 0.05 was considered to be statistically significant. This has been explicitly mentioned on Page no. 16; line no. 329-335.

Comment: Discussion should be reviewed and rewritten with existing literature. There is a lack of mechanism.

Response: We acknowledge the comment by the reviewer. The mechanism of Renogrit against Vancomycin induced nephrotoxicity has now been explicitly discussed in the discussion section. (Page no. 29; line no. 621-624). Also, the summary figure 7 contains the mechanisms observed. This has indeed infused better sense of clarity in the discussion section. We appreciate this advice very much.

Comment: It is recommended an English revision for the text. Some grammar errors should be corrected.

Response: We are grateful to the reviewer for the comment. The grammatic errors have now been rectified across the whole manuscript text.

Comment: Figures or Graphs should be clearer. They are not clear to the reader.

Response: We acknowledge the reviewer for the remark. The Figures with higher resolution have now been uploaded for the consideration.

---

## [Decision Letter · Decision Letter 1]

13 Sep 2023

PONE-D-23-14497R1Renogrit Attenuates Vancomycin-induced Nephrotoxicity in Human Renal Spheroids and in Sprague-Dawley Rats by Regulating Kidney Injury Biomarkers and Creatinine/Urea ClearancePLOS ONE

Dear Dr. Varshney,

Thank you for submitting your manuscript to PLOS ONE. After careful consideration, we feel that it has merit but does not fully meet PLOS ONE’s publication criteria as it currently stands. Therefore, we invite you to submit a revised version of the manuscript that addresses the points raised during the review process.

We look forward to receiving your revised manuscript.

Kind regards,

Ahmed E. Abdel Moneim

Academic Editor

PLOS ONE

Journal Requirements:

Reviewers' comments:

Reviewer's Responses to Questions

**Comments to the Author**

1. If the authors have adequately addressed your comments raised in a previous round of review and you feel that this manuscript is now acceptable for publication, you may indicate that here to bypass the “Comments to the Author” section, enter your conflict of interest statement in the “Confidential to Editor” section, and submit your "Accept" recommendation.

Reviewer #1: All comments have been addressed

Reviewer #3: All comments have been addressed

2. Is the manuscript technically sound, and do the data support the conclusions?

Reviewer #1: Yes

Reviewer #3: Yes

3. Has the statistical analysis been performed appropriately and rigorously? 

Reviewer #1: Yes

Reviewer #3: Yes

4. Have the authors made all data underlying the findings in their manuscript fully available?

Reviewer #1: Yes

Reviewer #3: Yes

5. Is the manuscript presented in an intelligible fashion and written in standard English?

Reviewer #1: Yes

Reviewer #3: Yes

6. Review Comments to the Author

Reviewer #1: The manuscript has been greatly improved based on the comments provided in the first revision. I have no further comments.

Reviewer #3: The authors have answered the questions. Unfortunately, the answered comments are not marked with a yellow bar in the main text of the article. This can help the reviewer to check.The reference for all methods is not mentioned.

7. PLOS authors have the option to publish the peer review history of their article (what does this mean?). If published, this will include your full peer review and any attached files.

Reviewer #1: No

Reviewer #3: No

---

## [Author Response · Author response to Decision Letter 1]

22 Sep 2023

Reviewer 1

Comment: The manuscript has been greatly improved based on the comments provided in the first revision. I have no further comments.

Response: We thank the reviewer for the comment.

Reviewer 3

Comment: The authors have answered the questions. Unfortunately, the answered comments are not marked with a yellow bar in the main text of the article. This can help the reviewer to check. The reference for all methods is not mentioned.

Response: We acknowledge the comment by the reviewer. The answered comments have now been marked with a yellow bar in the main text of the article. Also, additional references have been added in the methods section. As per the journal requirements, a clean copy of manuscript has to be provided, so highlighted version of the manuscript (with track changes) has been kept at the end of the clean copy of the manuscript.

---

## [Decision Letter · Decision Letter 2]

17 Oct 2023

Renogrit Attenuates Vancomycin-induced Nephrotoxicity in Human Renal Spheroids and in Sprague-Dawley Rats by Regulating Kidney Injury Biomarkers and Creatinine/Urea Clearance

PONE-D-23-14497R2

Dear Dr. Varshney,

We’re pleased to inform you that your manuscript has been judged scientifically suitable for publication and will be formally accepted for publication once it meets all outstanding technical requirements.

Kind regards,

Ahmed E. Abdel Moneim

Academic Editor

PLOS ONE

Additional Editor Comments (optional):

Reviewers' comments:

Reviewer's Responses to Questions

**Comments to the Author**

1. If the authors have adequately addressed your comments raised in a previous round of review and you feel that this manuscript is now acceptable for publication, you may indicate that here to bypass the “Comments to the Author” section, enter your conflict of interest statement in the “Confidential to Editor” section, and submit your "Accept" recommendation.

Reviewer #3: All comments have been addressed

2. Is the manuscript technically sound, and do the data support the conclusions?

Reviewer #3: Yes

3. Has the statistical analysis been performed appropriately and rigorously? 

Reviewer #3: Yes

4. Have the authors made all data underlying the findings in their manuscript fully available?

Reviewer #3: Yes

5. Is the manuscript presented in an intelligible fashion and written in standard English?

Reviewer #3: Yes

6. Review Comments to the Author

Reviewer #3: The article is acceptable. All questions have been answered. After checking again, there is no other comment.

7. PLOS authors have the option to publish the peer review history of their article (what does this mean?). If published, this will include your full peer review and any attached files.

Reviewer #3: No

---

## [Editor Report · Acceptance letter]

31 Oct 2023

PONE-D-23-14497R2 

Renogrit Attenuates Vancomycin-induced Nephrotoxicity in Human Renal Spheroids and in Sprague-Dawley Rats by Regulating Kidney Injury Biomarkers and Creatinine/Urea Clearance 

Dear Dr. Varshney:

I'm pleased to inform you that your manuscript has been deemed suitable for publication in PLOS ONE. Congratulations! Your manuscript is now with our production department. 

Kind regards, 

on behalf of

Dr. Ahmed E. Abdel Moneim 

Academic Editor

PLOS ONE